# ResiDual Transformer Alignment
# with Spectral Decomposition

**Lorenzo Basile**[*,†]                                                    *lorenzo.basile@phd.units.it*
*University of Trieste*

**Valentino Maiorca**[*,†]                                                    *maiorca@di.uniroma1.it*
*Sapienza University of Rome*
*Institute of Science and Technology Austria (ISTA)*

**Luca Bortolussi**                                                    *luca.bortolussi@units.it*
*University of Trieste*

**Emanuele Rodolà**                                                    *rodola@di.uniroma1.it*
*Sapienza University of Rome*

**Francesco Locatello**                                                    *francesco.locatello@ist.ac.at*
*Institute of Science and Technology Austria (ISTA)*

**Reviewed on OpenReview:** *https://openreview.net/forum?id=z37LCgSIzI*

## Abstract

When examined through the lens of their residual streams, a puzzling property emerges in transformer networks: residual contributions (e.g., attention heads) sometimes specialize in specific tasks or input attributes. In this paper, we analyze this phenomenon in vision transformers, focusing on the spectral geometry of residuals, and explore its implications for modality alignment in vision-language models. First, we link it to the intrinsically low-dimensional structure of visual head representations, zooming into their principal components and showing that they encode specialized roles across a wide variety of input data distributions. Then, we analyze the effect of head specialization in multimodal models, focusing on how improved alignment between text and specialized heads impacts zero-shot classification performance. This specialization-performance link consistently holds across diverse pre-training data, network sizes, and objectives, demonstrating a powerful new mechanism for boosting zero-shot classification through targeted alignment. Ultimately, we translate these insights into actionable terms by introducing ResiDual, a technique for spectral alignment of the residual stream. Much like panning for gold, it lets the noise from irrelevant unit principal components (i.e., attributes) wash away to amplify task-relevant ones. Remarkably, this dual perspective on modality alignment yields fine-tuning level performance on different data distributions while modelling an extremely interpretable and parameter-efficient transformation, as we extensively show on 70 pre-trained network-dataset combinations (7 models, 10 datasets).

## 1 Introduction

In recent times, transformers have become the backbone of most state-of-the-art machine learning systems, thanks to their adaptability to various domains, including language modeling (Brown et al., 2020; Touvron et al., 2023), vision (Dosovitskiy et al., 2021; Radford et al., 2021) and many different scientific domains (Espeholt et al., 2022; Jumper et al., 2021; Merchant et al., 2023). Traditionally, these models are treated as

---

*Equal contribution. [†] Work done while visiting ISTA. Code is available at `https://github.com/Flegyas/ResiDual`

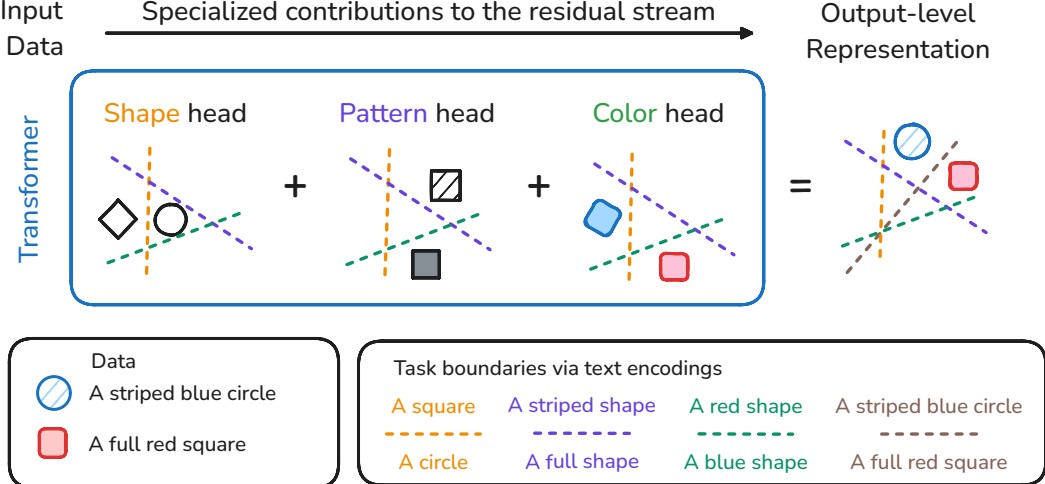

Figure 1: From the transformer's residual stream, direct contributions from individual heads across the network can be analyzed. In a multimodal, zero-shot classification setting (e.g., in CLIP), task boundaries are defined by text prompts that may vary in their conceptual granularity. When certain heads are specialized in particular features (e.g., shape, pattern, color), they may more accurately apply these boundaries than the model's original output. In this example, only the fine-grained task (brown) effectively separates the samples at the output level.

producing a unique, monolithic output. However, a key component in the success of transformers is the versatile inductive bias introduced by multi-head attention (MHA) layers, which alternate with multi-layer perceptrons (MLP) to form any transformer-based architecture. MHA layers are made of several independent computational units, called heads, that process input data in parallel and update the residual stream that carries it to the output via skip connections.

Similarly to what had been observed for filters of convolutional neural networks (Yosinski et al., 2014; Gavrikov & Keuper, 2022), recent works point to the emergence of a *specialization* property in attention heads, both in large language models (Voita et al., 2019; Li et al., 2023; Chughtai et al., 2024) and in the visual branch of CLIP models (Gandelsman et al., 2024). Specialization seems to be a by-product related to large-scale training, but it is not clear exactly why it emerges and whether this is a systematic property. Interestingly, this property implies that different units might learn to attend to specific attributes or to solve specific tasks, thus processing the input in a disentangled manner, overcoming known theoretical challenges (Hyvärinen & Pajunen, 1999; Locatello et al., 2019).

In modern transformer networks, the model's final output is produced by applying a simple linear transformation to the residual stream (up to LayerNorm). This residual stream accumulates information additively, drawing from each attention head and all MLP layers (Elhage et al., 2021), producing a general-purpose representation used as a feature set for many tasks. This decomposition raises an intriguing question: are all these units essential for solving specific tasks, or do some introduce noise that obscures task-relevant information? Typically, there is a trade-off between a model's generalization and its performance on specific tasks. However, it might be that to specialize a model for a specific task, we do not need to retrain the whole model. This is how we can define **task-irrelevant** and **noise** units: task-irrelevant units are those that can be deactivated without affecting performance, while noise refers to units that, when deactivated, enhance performance by removing signals misaligned with the task. By manipulating the residual stream, we can boost the units already aligned with the task, amplifying relevant signals while reducing noise.

**Contribution** In this paper, we tackle this question from the perspective of the latent geometry of residual units. First, on a variety of transformer-based vision models, including multiple versions of CLIP (Radford et al., 2021), BLIP (Li et al., 2022), ViT (Dosovitskiy et al., 2021) and DINOv2 (Oquab et al., 2024), we

show that such units are embedded in low-dimensional manifolds and that, when there is specialization, it can be traced back to the role of few principal components. Then, by introducing a spectral analysis method based on a discrete version of principal angles (Björck & Golub, 1973), we quantitatively measure the similarity of residual units across different datasets, revealing that the roles of specialized units remain surprisingly stable.

Building on this insight, we hypothesize that, in many cases, the information necessary for solving a task is already embedded within a subset of highly specialized residual units. We show that this picture emerges clearly in vision-language models like CLIP, where we find units or sets of units that align with textual attributes more precisely than the full model output on a given task. In fact, as the output combines all residual units, this relevant information may be obfuscated by other units that introduce irrelevant signals. Instead of fine-tuning the entire model, we propose to isolate and enhance the task-relevant units by filtering out the noise – akin to panning for gold. By doing so, we can significantly boost model performance with up to 4 orders of magnitude fewer parameters than full fine-tuning and 2 less than those needed for training a simple linear transformation at the output level.

To implement this, we introduce *ResiDual*, a novel approach that focuses on the principal components (PCs) of the residual units to identify and retain the needed information. This framework selectively reweights the most relevant PCs, amplifying the signals that align with the task objective while remaining computationally efficient. This spectral reweighting of individual units addresses nonlinear interactions between them and provides a geometrically principled and interpretable method for optimizing transformer models by capitalizing on the knowledge they already possess.

In summary, our contributions are as follows:

- We inspect the geometric structure of attention head representations in vision transformers, showcasing their low dimensionality and their increasing nonlinearity along model depth;

- We characterize the emergent specialization of attention heads through their principal components and show that it stays consistent across data distributions;

- We identify task-specific units in vision-language models, showcasing that focusing on these units in zero-shot settings can outperform using the full residual output when there is latent alignment between units and tasks;

- We introduce *ResiDual*, a geometrically grounded method for manipulating transformers by reweighting the most relevant principal components of the residual units. This approach can sidestep the need for full-model finetuning, as it reaches competitive performance with minimal parameter overhead.

## 2 Related Work

**Transformer Residual Decomposition**  Transformer networks (Vaswani et al., 2017) rely on residual connections around each multi-head attention (MHA) and MLP layer, resulting in a final representation that combines contributions from all units across layers by simple summation. Techniques like logit lens (nostalgebraist, 2020) and Direct Logit Attribution (DLA) (Elhage et al., 2021) – and Cancedda (2024), at the spectral level – focus on how individual layers or residual units (such as MLPs or attention heads) affect the final output in logit space, given that their contributions are projected upstream via linear transformations (up to LayerNorm (Lei Ba et al., 2016), an affine one). Here, we apply this residual decomposition to provide a more comprehensive understanding of how these units interact and align across different tasks, revealing the deeper structure within the residual space.

**Residual Properties**  To understand the geometric nature of latent manifolds, previous works analyze the intrinsic dimensionality ($I_d$) (Ansuini et al., 2019; Cheng et al., 2023; Valeriani et al., 2024) of the representations within the network, which is typically much lower than the embedding dimension. We posit that the Linear Representation Hypothesis (Park et al.; Jiang et al., 2024), which suggests that transformer representations encode high-level concepts in linear directions, complements this view hinting at transformer

models sometimes modulating input attributes in a linearly structured and low-dimensional (i.e., specialized) way. Previous works in language modeling have highlighted this specialization (Voita et al., 2019; Michel et al., 2019; Li et al., 2023; Lv et al., 2024; Chughtai et al., 2024), revealing that only a few attention heads are responsible for specific tasks and that they assume specialized and interpretable roles. Our analysis bridges geometry and specialization, revealing that vision transformer heads are low-dimensional (though often nonlinear, especially in deeper layers) and highly specialized for downstream tasks.

**Multimodal Alignment** In multimodal models such as CLIP (Radford et al., 2021), it is well known that the vision and text branches operate in neatly separated latent spaces (Liang et al., 2022). Despite this modality gap, Chattopadhyay et al. (2024) and Bhalla et al. (2024) leverage the multimodal latent space of CLIP to find sparse decompositions of its representations using text. Similarly, Gandelsman et al. (2024) show that text encodings can align with specific head-level representations of CLIP's visual branch, providing insights into the specialized roles of individual heads through manually crafted textual inputs. Balasubramanian et al. (2024) generalize this approach to unimodal vision transformers and arbitrary residual units through a scoring function and the estimation of an aligning transformation between the spaces. The idea that CLIP is able to disentangle concepts and encode them in separate subspaces also appears in Wolff et al. (2023) and Lewis et al. (2024). Here, we study the modality alignment at the spectral level, reaching the granularity of head principal components, and show how they can be used to improve the alignment between text and visual branches in CLIP-like models.

## 3 The Geometry of Residual Units

In this section, we examine the relationship between head specialization and the low-dimensional nature of head manifolds. Initially, head representations exist in a relatively low-dimensional ambient space. Through a linear transformation, however, they are subsequently embedded into a higher-dimensional space within the residual stream (Elhage et al., 2021), which shares the same dimensionality as the model's output. At this point, head representations are transcribed to the residual stream, and they contribute additively to the final output of the model. In fact, throughout the paper, we will assume that the model output is the summation of the encodings of all residual units (attention heads $\boldsymbol{H}$, MLPs $\boldsymbol{M}$ and input embeddings $\boldsymbol{X}_0$):

$$\boldsymbol{Y} = \sum_{i=1}^{|\mathbb{U}|} \boldsymbol{U}_i = \boldsymbol{X}_0 + \sum_{i=1}^{|\mathbb{H}|} \boldsymbol{H}_i + \sum_{j=1}^{|\mathbb{M}|} \boldsymbol{M}_j \,, \tag{1}$$

with $\boldsymbol{Y}$ as the final output of the model, summing up all the residual units in $\boldsymbol{U} \in \mathbb{U}$. Please refer to Appendix A.1 for a more rigorous description of the residual decomposition. In this paper, we focus on the geometry of attention head representations, as prior works (e.g., Voita et al. (2019); Gandelsman et al. (2024)) have demonstrated their specialization in distinct tasks. MLP layers mix outputs from multiple heads by design, leading to more entangled and less interpretable representations (as we show in Figure 11).

### 3.1 Residual Dimensionality

Despite being embedded into a higher-dimensional space, head representations exhibit an even lower *intrinsic dimensionality* ($I_d$) than that of the original ambient space. This indicates that irrespective of their high-dimensional embedding, the essential structure of head representations is highly compressed and governed by a compact, low-dimensional geometry. In short, the intrinsic dimensionality of a dataset is the least number of variables required to satisfactorily describe the data points. Ideally, if data lie on a linear manifold (a hyperplane), the intrinsic dimensionality coincides with the number of principal components required to completely explain their variance. In a more realistic setting, data lie on curved, nonlinear manifolds, and linear estimators like PCA fail to capture their real intrinsic dimensionality. In such cases, one can resort to nonlinear $I_d$ estimators. Among them, we choose to employ the TwoNN (Facco et al., 2017) because of its efficiency and stability on complex and non-uniform manifolds.

**Experimental setting** We start by evaluating the intrinsic dimensionality of head representations across multiple transformer-based vision architectures pre-trained with different objectives (supervised, unsuper-

vised, self-supervised). Namely, we employ OpenAI's CLIP (Radford et al., 2021), OpenCLIP (Cherti et al., 2023), BLIP (Li et al., 2022), ViT (Dosovitskiy et al., 2021) and DINOv2 (Oquab et al., 2024), all in their version based on ViT-Large (results on ViT-Base models are in the Appendix in Figure 8). We feed them a subset of the training set of ImageNet (Russakovsky et al., 2015) containing 80000 images stratified on the class labels, and we extract the representations for all attention heads. Then, we compute the intrinsic dimensionality of such representations using a linear estimator (PCA) and a nonlinear one (TwoNN). Linear $I_d$ is computed as the number of components needed by PCA to explain 99% of head variance.

| Layer | OpenCLIP-L L | N | Ratio | EVR$_1$ | CLIP-L L | N | Ratio | EVR$_1$ | DINOv2-L L | N | Ratio | EVR$_1$ | ViT-L L | N | Ratio | EVR$_1$ | BLIP-L L | N | Ratio | EVR$_1$ |
|---|---|---|---|---|---|---|---|---|---|---|---|---|---|---|---|---|---|---|---|---|
| 23 | 60.19 | 19.14 | 3.19 | 0.09 | 60.38 | 20.26 | 3.02 | 0.09 | 62.56 | 14.61 | 4.28 | 0.08 | 61.56 | 24.20 | 2.56 | 0.11 | 50.12 | 15.06 | 3.36 | 0.14 |
| 22 | 61.00 | 20.61 | 3.00 | 0.11 | 61.81 | 21.05 | 2.95 | 0.08 | 62.44 | 16.99 | 3.69 | 0.08 | 62.06 | 28.33 | 2.20 | 0.08 | 50.06 | 14.00 | 3.60 | 0.16 |
| 21 | 61.94 | 23.15 | 2.70 | 0.10 | 62.00 | 21.14 | 2.95 | 0.10 | 62.06 | 18.37 | 3.39 | 0.09 | 62.00 | 30.45 | 2.05 | 0.07 | 49.19 | 15.93 | 3.13 | 0.27 |
| 20 | 61.88 | 25.71 | 2.42 | 0.09 | 62.00 | 22.47 | 2.78 | 0.08 | 61.81 | 20.41 | 3.04 | 0.11 | 62.19 | 32.57 | 1.91 | 0.06 | 53.75 | 20.02 | 2.69 | 0.22 |
| 19 | 62.25 | 25.94 | 2.41 | 0.09 | 61.81 | 22.95 | 2.71 | 0.10 | 61.06 | 21.81 | 2.82 | 0.15 | 61.88 | 33.87 | 1.83 | 0.05 | 56.50 | 22.76 | 2.49 | 0.18 |
| 18 | 62.00 | 26.23 | 2.38 | 0.09 | 61.94 | 25.08 | 2.50 | 0.11 | 60.12 | 24.22 | 2.49 | 0.16 | 61.81 | 34.36 | 1.80 | 0.05 | 56.94 | 23.91 | 2.38 | 0.14 |
| 17 | 61.94 | 27.46 | 2.29 | 0.09 | 61.88 | 26.46 | 2.36 | 0.09 | 60.06 | 24.56 | 2.47 | 0.15 | 61.44 | 33.63 | 1.83 | 0.05 | 57.50 | 24.49 | 2.35 | 0.13 |
| 16 | 61.69 | 28.59 | 2.17 | 0.08 | 61.50 | 28.26 | 2.19 | 0.12 | 60.19 | 25.92 | 2.33 | 0.16 | 61.81 | 34.46 | 1.80 | 0.05 | 56.31 | 24.23 | 2.33 | 0.14 |
| 15 | 61.75 | 29.82 | 2.08 | 0.09 | 61.44 | 28.75 | 2.15 | 0.12 | 59.38 | 26.49 | 2.26 | 0.16 | 60.81 | 33.48 | 1.82 | 0.06 | 56.12 | 24.69 | 2.28 | 0.15 |
| 14 | 61.75 | 30.49 | 2.03 | 0.11 | 61.62 | 30.40 | 2.03 | 0.14 | 59.38 | 27.42 | 2.17 | 0.17 | 60.88 | 33.78 | 1.80 | 0.05 | 55.50 | 24.74 | 2.25 | 0.15 |
| 13 | 61.00 | 29.08 | 2.11 | 0.14 | 61.38 | 30.31 | 2.03 | 0.13 | 57.50 | 26.65 | 2.16 | 0.21 | 60.00 | 33.83 | 1.78 | 0.07 | 54.75 | 25.13 | 2.18 | 0.14 |
| 12 | 60.94 | 30.28 | 2.02 | 0.16 | 60.50 | 30.21 | 2.01 | 0.13 | 57.00 | 26.28 | 2.17 | 0.18 | 58.44 | 33.92 | 1.73 | 0.08 | 53.88 | 24.83 | 2.17 | 0.16 |
| 11 | 60.31 | 29.66 | 2.04 | 0.15 | 60.56 | 30.61 | 1.98 | 0.14 | 56.25 | 27.26 | 2.07 | 0.22 | 55.19 | 32.43 | 1.70 | 0.09 | 48.81 | 23.14 | 2.11 | 0.18 |
| 10 | 59.69 | 29.63 | 2.02 | 0.14 | 60.50 | 31.22 | 1.94 | 0.10 | 54.75 | 26.49 | 2.07 | 0.24 | 51.38 | 31.07 | 1.65 | 0.10 | 43.38 | 21.66 | 2.00 | 0.22 |
| 9 | 59.56 | 30.34 | 1.97 | 0.14 | 58.69 | 29.13 | 2.02 | 0.15 | 45.50 | 22.47 | 2.02 | 0.29 | 48.94 | 30.19 | 1.62 | 0.10 | 42.81 | 21.50 | 1.98 | 0.23 |
| 8 | 57.56 | 28.06 | 2.05 | 0.15 | 58.44 | 28.58 | 2.05 | 0.15 | 43.69 | 22.36 | 1.92 | 0.42 | 45.56 | 28.65 | 1.59 | 0.14 | 38.88 | 20.50 | 1.89 | 0.26 |
| 7 | 55.31 | 26.96 | 2.05 | 0.18 | 57.19 | 27.91 | 2.05 | 0.17 | 47.50 | 24.10 | 1.94 | 0.27 | 44.81 | 27.62 | 1.62 | 0.15 | 34.94 | 19.24 | 1.81 | 0.30 |
| 6 | 51.75 | 24.91 | 2.08 | 0.25 | 53.19 | 25.48 | 2.09 | 0.21 | 41.94 | 21.28 | 1.94 | 0.35 | 43.50 | 27.29 | 1.59 | 0.14 | 29.19 | 17.56 | 1.65 | 0.33 |
| 5 | 48.44 | 23.99 | 2.01 | 0.25 | 50.56 | 24.51 | 2.06 | 0.28 | 47.38 | 23.89 | 1.97 | 0.31 | 40.62 | 25.22 | 1.61 | 0.16 | 32.38 | 18.51 | 1.74 | 0.32 |
| 4 | 38.81 | 20.37 | 1.88 | 0.36 | 49.62 | 24.35 | 2.04 | 0.31 | 32.81 | 18.36 | 1.75 | 0.38 | 39.81 | 24.58 | 1.62 | 0.20 | 25.75 | 16.10 | 1.57 | 0.35 |
| 3 | 30.19 | 18.30 | 1.62 | 0.37 | 45.38 | 22.62 | 1.98 | 0.30 | 42.25 | 23.20 | 1.78 | 0.53 | 36.94 | 22.52 | 1.64 | 0.24 | 28.62 | 17.04 | 1.67 | 0.30 |
| 2 | 29.50 | 19.76 | 1.40 | 0.54 | 47.56 | 24.24 | 1.92 | 0.24 | 29.94 | 17.66 | 1.61 | 0.43 | 31.75 | 19.30 | 1.64 | 0.25 | 25.06 | 16.09 | 1.54 | 0.38 |
| 1 | 31.38 | 19.28 | 1.51 | 0.49 | 45.88 | 23.81 | 1.88 | 0.28 | 12.69 | 11.65 | 1.06 | 0.56 | 23.25 | 15.57 | 1.45 | 0.35 | 18.25 | 14.06 | 1.26 | 0.49 |
| 0 | 33.44 | 20.16 | 1.52 | 0.50 | 42.44 | 22.53 | 1.80 | 0.39 | 11.81 | 11.45 | 0.98 | 0.59 | 15.00 | 13.11 | 1.03 | 0.56 | 11.81 | 13.54 | 0.80 | 0.66 |

Figure 2: Heads in early layers show low-dimensional, linear structures, as suggested by similar intrinsic dimension estimates from PCA (L) and TwoNN (N). Moving toward the output layer, the nonlinear dimensionality peaks and then decreases, while PCA's linear estimate continues to rise, indicating increasing nonlinearity in head manifolds (Ratio $= \frac{L}{N}$). The first principal component (EVR$_1$) explains around 50% of the variance in early layers, dropping to around 10% in later layers.

**Result analysis** We report in Figure 2 the results for head units on ImageNet (a subset of 80000 images stratified by class), in Figure 10 the results for other datasets and in Figure 11 those for MLP units on ImageNet. We observe that the *true* head dimensionality (the one computed with a nonlinear estimator, TwoNN) tends to increase in the first half of the model and to decrease towards the last few layers, following a characteristic hunchback shape, similar to previous findings in other vision architectures (Ansuini et al., 2019). However, the number of dimensions returned by the linear estimator grows constantly through the model. This disparity, reflected in the growing ratio between the two estimates, suggests that the units in the early layers are close to linear, while those in the later layers lie on more curved manifolds. The last column shows the average explained variance ratio (EVR) of the first PCA component and highlights that heads in the first layers are largely explained by this direction, while it still accounts for a nontrivial 10% of head variance in late layers.

These findings highlight that low head dimensionality and monotonically increasing nonlinearity arise along the residual streams of vision transformers, regardless of pre-training objective and data.

## 3.2 Principal Components Encode Unit Semantics

The low dimensionality of head encodings results in relevant consequences for their interpretability. Head representations can be easily approximated with sparse recovery algorithms in a way that is akin to performing PCA, but over a discrete set of vectors. For CLIP models, this approach has been recently explored by

Gandelsman et al. (2024). There, the authors introduce a sparse approximation algorithm, TextSpan (TS), and decompose head encodings using a set of textual descriptions coming from the text branch of CLIP as a dictionary. They observe strong specialization properties, witnessed by high coherence in the textual explanations of each head.

We link TextSpan to the more established family of Matching Pursuit (MP) (Mallat & Zhang, 1993) algorithms, widely employed in signal processing. More specifically, as we show in the Appendix A.2, TextSpan is analogous to Simultaneous Orthogonal Matching Pursuit (SOMP) (Tropp et al., 2006), with light modifications. TextSpan, like any MP algorithm, approximates the signal through linear combinations of basis functions. Considering the high nonlinearity of later layers (Section 3.1), and TS being a linear sparse approximation method, we now want to investigate whether TS is, in reality, focusing on the first principal components of the signal (head-level representations).

**Experimental setting**  For this experiment, we position ourselves in the same setup as the original TS paper (Gandelsman et al., 2024). Hence, we consider the attention heads belonging to the last 4 layers of OpenCLIP-L. We use two sparse approximation algorithms: the original TextSpan, which operates on the whole head representation, and Orthogonal Matching Pursuit (OMP) (Pati et al., 1993). Different from TextSpan, OMP computes sparse approximations of vectors, not matrices (like head representations). Therefore, in this experiment, we apply OMP to the first principal component of each head. We denote this method as $OMP_1$. The dictionary we use contains the encodings produced by OpenCLIP-L for the set of image descriptions provided by Gandelsman et al. (2024). We apply both algorithms to select 5 descriptions for each head and compute an agreement score between the two sets. The agreement is computed as the absolute Z-score of the cosine similarity ($\texttt{sim}$) between them, compared with the average cosine similarity $\mu$ between the descriptions selected by TextSpan and the entire dictionary $Z = \frac{|\texttt{sim}(\text{TS},\text{OMP}_1)-\mu|}{\sigma}$.

**Result analysis**  We report in the left panel of Figure 3 the agreement scores. The right panel reports a few examples (one per layer) of descriptions obtained using the two algorithms. We observe that a high Z-score (e.g., head 8 of layer 22, which is almost $5\sigma$ away from $\mu$), is reflected in extremely similar descriptions from the two methods. When the agreement is lower, as in the case of head 20 from layer 8, the two sets of descriptions substantially differ even though they share some high-level semantics.

Overall, this analysis indicates that, in some cases, the first principal component captures nearly all the essential information about the head's specialized semantics. In other cases, the head's role appears to be distributed across multiple components.

## 3.3   Spectral Dataset Comparison

Our aim is now to understand to what extent specialization generalizes across different input data distributions. To do so, we introduce a spectral metric to compare the representations of residual units. Since units are low-dimensional (Section 3.1) and their specialization is deeply impacted by a few principal components (Section 3.2), we define a metric to quantify the similarity between their PCA bases, inspired by principal angles (Björck & Golub, 1973).

Let $\mathcal{S}_1$ and $\mathcal{S}_2$ be two subspaces of dimensions $k_1$ and $k_2$ in an $d$-dimensional space. The principal angles $\theta_n$ for $n = 1, \dots, \min(k_1, k_2)$ are given by:

$$\cos \theta_n = \max_{\boldsymbol{u} \in \mathcal{S}_1^{\perp \mathcal{U}_{n-1}}, \boldsymbol{v} \in \mathcal{S}_2^{\perp \mathcal{V}_{n-1}}} \frac{\boldsymbol{u}^\top \boldsymbol{v}}{\|\boldsymbol{u}\|\|\boldsymbol{v}\|} \tag{2}$$

where $\mathcal{U}_{n-1}$ and $\mathcal{V}_{n-1}$ are the span of $\{\boldsymbol{u}_1, \dots, \boldsymbol{u}_{n-1}\}$ and $\{\boldsymbol{v}_1, \dots, \boldsymbol{v}_{n-1}\}$, respectively.

Now, let $\mathbb{S}_1 = \{\boldsymbol{u}_1, \dots, \boldsymbol{u}_{k_1}\}$ and $\mathbb{S}_2 = \{\boldsymbol{v}_1, \dots, \boldsymbol{v}_{k_2}\}$ represent sets of $\ell_2$-normalized discrete vectors (e.g., principal components) with (optional) associated weights $w_1^i$ and $w_2^j$ (e.g., singular values).

We define the **spectral cosine similarity** $s_n$ for $n = 1, \dots, \min(k_1, k_2)$ as:

$$s_n = [\max_{\substack{i \notin \{i_1, \dots, i_{n-1}\} \\ j \notin \{j_1, \dots, j_{n-1}\}}} (\boldsymbol{u}_i^\top \boldsymbol{v}_j)] w_1^i w_2^j \tag{3}$$

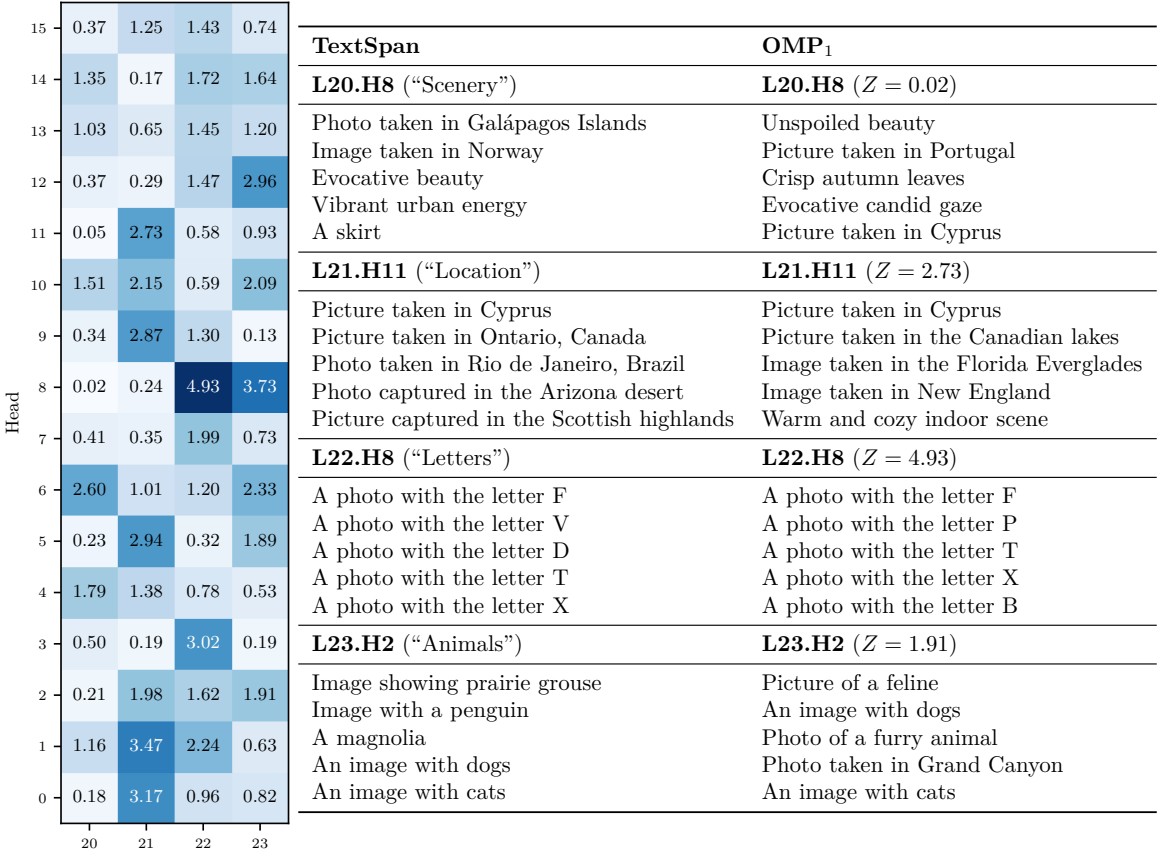

| TextSpan | OMP$_1$ |
|---|---|
| **L20.H8** ("Scenery") | **L20.H8** ($Z = 0.02$) |
| Photo taken in Galápagos Islands | Unspoiled beauty |
| Image taken in Norway | Picture taken in Portugal |
| Evocative beauty | Crisp autumn leaves |
| Vibrant urban energy | Evocative candid gaze |
| A skirt | Picture taken in Cyprus |
| **L21.H11** ("Location") | **L21.H11** ($Z = 2.73$) |
| Picture taken in Cyprus | Picture taken in Cyprus |
| Picture taken in Ontario, Canada | Picture taken in the Canadian lakes |
| Photo taken in Rio de Janeiro, Brazil | Image taken in the Florida Everglades |
| Photo captured in the Arizona desert | Image taken in New England |
| Picture captured in the Scottish highlands | Warm and cozy indoor scene |
| **L22.H8** ("Letters") | **L22.H8** ($Z = 4.93$) |
| A photo with the letter F | A photo with the letter F |
| A photo with the letter V | A photo with the letter P |
| A photo with the letter D | A photo with the letter T |
| A photo with the letter T | A photo with the letter X |
| A photo with the letter X | A photo with the letter B |
| **L23.H2** ("Animals") | **L23.H2** ($Z = 1.91$) |
| Image showing prairie grouse | Picture of a feline |
| Image with a penguin | An image with dogs |
| A magnolia | Photo of a furry animal |
| An image with dogs | Photo taken in Grand Canyon |
| An image with cats | An image with cats |

Figure 3: Comparison between TextSpan and Orthogonal Matching Pursuit on the first principal component (OMP$_1$), applied to the heads of OpenCLIP-L. *Left*: agreement score between the descriptions returned by the two methods. *Right*: qualitative comparison of selected descriptions for 4 heads, one per layer, at different agreement levels. A similar analysis for the second principal component is presented in the Appendix in Figure 9.

where $i_1, \ldots, i_{n-1}$ and $j_1, \ldots, j_{n-1}$ are previously selected indices.

The original principal angles measure the alignment between **subspaces** by maximizing the cosine similarity of vectors in their **span**. In our discrete case, vectors are directly selected from sets $\mathbb{S}_1$ and $\mathbb{S}_2$ with optional weighting. The final measure is the aggregation of the spectral cosine similarities along the $\min(k_1, k_2)$ entries, with normalization to bound our measure between 0 and 1. We define the **normalized spectral cosine similarity** between the two sets of principal components as:

$$\text{sim}(\mathbb{S}_1, \mathbb{S}_2) = \sqrt{\frac{\sum_{n=1}^{\min(k_1,k_2)} s_n^2}{\sum_{n=1}^{\min(k_1,k_2)} (w_1^n w_2^n)^2}} \tag{4}$$

In the following, we apply this measure to compare residual units across different input datasets. It is worth noting here that the main advantage of this formulation, compared to standard approaches to representation similarity, is that our metric does not rely on the alignment between samples, as it operates in the dual spectral space. This edge is crucial in our application to different datasets, which even vary in size.

**Experimental setting**  We consider the same ViT-based encoders of Section 3.1, and 14 different datasets: ImageNet (the same split used in Section 3.1), CIFAR(-100/-10) (Krizhevsky, 2009), ImageNet-Sketch (Wang

et al., 2019), Cars (Krause et al., 2013), MNIST (LeCun et al., 1998), SVHN (Netzer et al., 2011), EuroSAT (Helber et al., 2019), RESISC45 (Cheng et al., 2017), DTD (Cimpoi et al., 2014), SUN397 (Xiao et al., 2016), GTSRB (Stallkamp et al., 2011), PACS (Li et al., 2017), and random images (10000 samples with RGB values in $[-1, 1]$). We use the original train/validation/test splits if available, otherwise we produce the splits through a stratified random sampling over the classes. For each encoder, we use our similarity measure to compare its unit representations produced on each training dataset with the ones obtained on the training split of ImageNet. ImageNet is taken as a reference under the assumption that being a general enough dataset, its head PCA bases are sufficiently comprehensive to approximate the primary features across other datasets. Additionally, we perform a qualitative inspection of a few heads that stand out by finding their textual decomposition. For this step, we use Simultaneous Orthogonal Matching Pursuit (SOMP), having established its strong relationship with TextSpan (Appendix A.2).

**Result analysis** The results of the spectral head-to-head comparison between ImageNet and all other datasets on OpenCLIP-L are reported in Figure 4 (results on other models can be found in the Appendix in appendix A.4). Rows are ordered according to the mean overall similarity between the corresponding dataset and ImageNet. Interestingly, dataset ordering is consistent across different encoders. The mean correlation coefficient between dataset similarities (averaged over all heads) across different models is 0.97. The full comparison between encoders is reported in the Appendix (Figure 13). On OpenCLIP-L, we observe that datasets that maximally align with ImageNet share classes with it (e.g., SUN397 and Sketch) and/or contain generic images (e.g., DTD and Cars). Moreover, datasets that share the same input image structure and concepts (CIFAR-10 and CIFAR-100) have an almost identical similarity distribution across heads. Overall, we observe a decreasing trend in head similarity scores as depth in the model increases. The simple, linear heads of the first layers are responsible for the extraction of low-level patterns (Dosovitskiy et al., 2021) and emerge as almost always identical across different data distributions. On the last layers, just a few heads per dataset stand out: this is where we are looking for specialization. Zooming in on a few of these heads, in Figure 4b, we report their textual descriptions obtained with SOMP. Head 7 of layer 22 (specialized on seasons) stands out because it is highly activated in many datasets that contain pictures of scenery (such as SUN397, GTSRB, and, more prominently, EuroSAT). Head 11 of layer 22 (specialized in shades of gray) emerges as extremely different between ImageNet and Sketch, which contains grayscale drawings of ImageNet classes. Head 10 of layer 23 (specialized in numbers) is highly activated on both MNIST and SVHN. We note that this is not the only 'shared' head between the two, but others, like head 1 of the same layer, are also activated by the random dataset, signaling that they are not as specific.

These findings show that the similarity measure yields intuitive scores, with ImageNet's foundational attributes demonstrating generalizability across various data distributions.

# 4  ResiDual Alignment

With a refined understanding of head specialization, our objective is now to leverage this property to enhance alignment between visual unit representations (both heads and MLPs) and text encodings in CLIP-like models. From this point onwards, our experiments will have a shared objective: given a frozen zero-shot classifier, e.g., text encodings or prototypes for classes, we manipulate the residual to improve its alignment with this target subspace. In the main text, we focus on multimodal architectures, where improved alignment directly benefits tasks such as zero-shot classification by strengthening the model's capacity to interpret visual data through text-based descriptors. Additionally, in Appendix A.3, we conduct experiments on unimodal encoders, where the zero-shot classifier is built in a prototype-based fashion.

## 4.1  Coarse Unit Alignment

We start by exploring whether certain heads are already aligned with the text subspace relevant to our task. Specifically, our goal is to optimize vision-text alignment by combining visual heads weighted using various scoring functions. These functions assign weights to entire head representations to modulate (in some cases completely remove) their contributions to the residual stream. Crucially, these methods operate exclusively

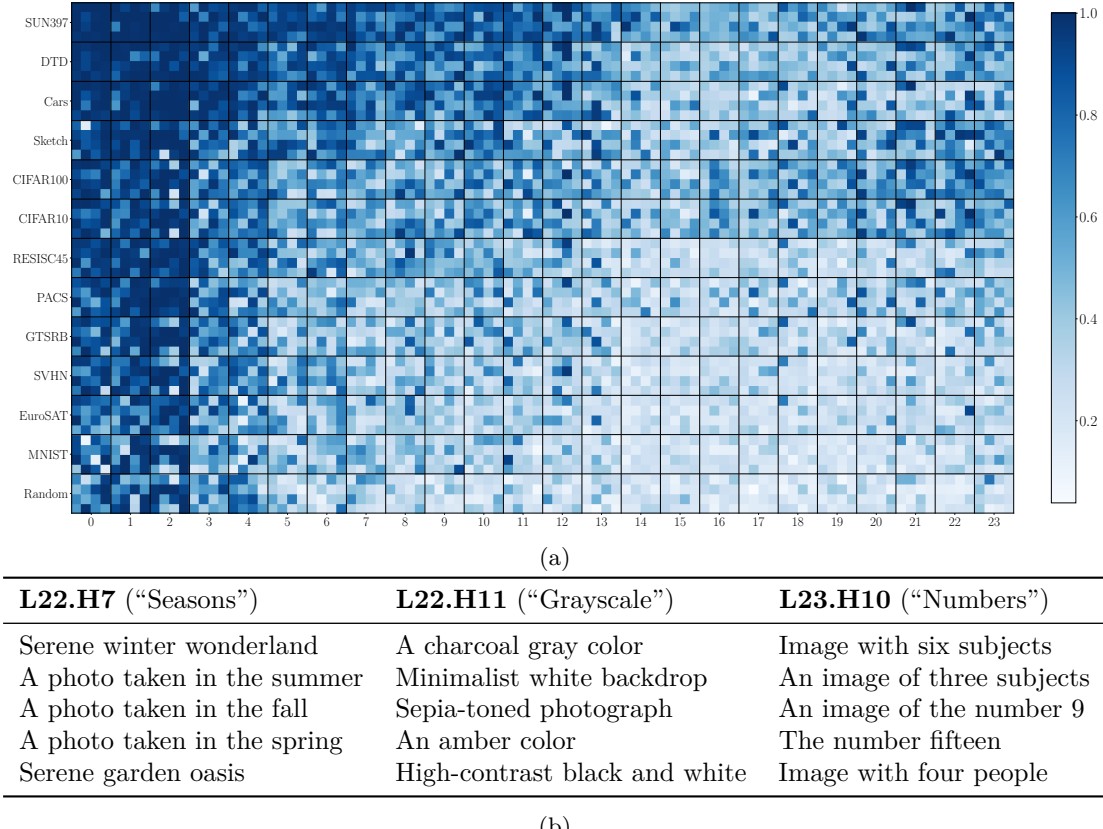

(a)

| **L22.H7** ("Seasons") | **L22.H11** ("Grayscale") | **L23.H10** ("Numbers") |
| --- | --- | --- |
| Serene winter wonderland | A charcoal gray color | Image with six subjects |
| A photo taken in the summer | Minimalist white backdrop | An image of three subjects |
| A photo taken in the fall | Sepia-toned photograph | An image of the number 9 |
| A photo taken in the spring | An amber color | The number fifteen |
| Serene garden oasis | High-contrast black and white | Image with four people |

(b)

Figure 4: *(a)* Attention head similarity across layers of OpenCLIP-L, computed between ImageNet head representations and those obtained on other datasets. *(b)* Descriptions picked by SOMP for three specialized heads that emerge from the analysis of panel *(a)*.

at the level of whole-head reweighting, leaving the internal structure and intrinsic specialization of individual heads unchanged. As a result, only the output specialization is affected.

**Experimental setting**   We have 3 different selection methods: i) **Unsupervised (U):** We use the head-to-output correlation as a measure. Intuitively, the more one head correlates to the full output, the more information it carries. In practice, we compute the Pearson correlation between each sample at the head level and its corresponding output encoding, averaging across samples to obtain a scalar; ii) **Task-conditioned Unsupervised (U|T)**: Conditioning on the output alone does not necessarily imply that the selected heads will be suitable for a given task. Since the task is modeled by a text subspace, having one encoding for each class, we can condition the previous unsupervised measure to be applied only on the head and output subspaces spanned by the task encodings. This has the effect of ignoring features that might influence the correlation but are not related to the task at hand. This is a direct application of the CompAttribute metric introduced in Balasubramanian et al. (2024); iii) **Supervised (S):** When we assume the availability not only of the task encodings but also of labeled samples, we can directly estimate the head score by looking at its performance on the downstream task (in the style of logit lens (nostalgebraist, 2020)).

For each scoring function, we evaluate each head individually, rank them according to their scores, and apply a greedy top-k selection. We then sum these selected heads $\mathbb{H}' \subseteq \mathbb{H}$ to create a partially recovered residual, which is subsequently evaluated on downstream task performance:

$$\boldsymbol{Y}' = \sum_{i=1}^{|\mathbb{H}'|} \boldsymbol{H}_i \,. \tag{5}$$

| | BLIP-L | | | | | | | OpenCLIP-L | | | | | | |
| Dataset | U | U\|T | S | R | H | B | O | U | U\|T | S | R | H | B | O |
|---|---|---|---|---|---|---|---|---|---|---|---|---|---|---|
| CIFAR10 | 0.94 | 0.94 | 0.93 | 0.56 | 0.93 | 0.94 | 0.96 | 0.96 | 0.96 | 0.96 | 0.59 | 0.94 | 0.97 | 0.98 |
| CIFAR100 | 0.70 | 0.71 | 0.72 | 0.33 | 0.69 | 0.71 | 0.75 | 0.80 | 0.78 | 0.79 | 0.38 | 0.76 | 0.82 | 0.84 |
| Cars | 0.67 | 0.68 | 0.71 | 0.27 | 0.66 | 0.72 | 0.77 | 0.92 | 0.92 | 0.93 | 0.39 | 0.89 | 0.93 | 0.93 |
| DTD | 0.52 | 0.54 | 0.54 | 0.27 | 0.50 | 0.55 | 0.61 | 0.59 | 0.59 | 0.59 | 0.29 | 0.54 | 0.63 | 0.69 |
| EuroSAT | 0.49 | 0.53 | 0.53 | 0.24 | 0.37 | 0.50 | 0.92 | 0.64 | 0.64 | 0.67 | 0.34 | 0.55 | 0.64 | 0.95 |
| GTSRB | 0.34 | 0.34 | 0.37 | 0.14 | 0.33 | 0.35 | 0.59 | 0.55 | 0.56 | 0.55 | 0.20 | 0.49 | 0.56 | 0.75 |
| MNIST | 0.62 | 0.64 | 0.65 | 0.27 | 0.41 | 0.52 | 0.94 | 0.74 | 0.84 | 0.85 | 0.25 | 0.41 | 0.54 | 0.97 |
| RESISC45 | 0.58 | 0.57 | 0.60 | 0.28 | 0.56 | 0.59 | 0.79 | 0.70 | 0.69 | 0.70 | 0.33 | 0.63 | 0.73 | 0.86 |
| SUN397 | 0.67 | 0.66 | 0.68 | 0.28 | 0.65 | 0.70 | 0.73 | 0.70 | 0.69 | 0.71 | 0.27 | 0.59 | 0.74 | 0.76 |
| SVHN | 0.25 | 0.36 | 0.42 | 0.16 | 0.21 | 0.33 | 0.56 | 0.48 | 0.57 | 0.53 | 0.24 | 0.41 | 0.41 | 0.70 |
| **Average** | 0.58 | 0.60 | **0.61** | 0.28 | 0.53 | 0.59 | 0.76 | 0.71 | **0.73** | **0.73** | 0.33 | 0.62 | 0.70 | 0.84 |

Table 1: Accuracy when doing **zero ablation of all units except top 5%** of attention heads. Heads are assigned a binary weight using an Unsupervised (U), Task-conditioned Unsupervised (U|T), Supervised (S), and Random (R) strategy (a mean over 10 different seeds). H corresponds to using all the attention heads available, B is the original model performance, and O is the optimized continuous weighting case. For a qualitative analysis complementing these quantitative results on a selection of datasets, please refer to Table 6 in the Appendix.

We have 3 control measures in place: i) **Heads (H):** The performance of the model when *all* the attention heads, and only them, are used. This gives information about the heads' contribution to the residual and, symmetrically, how much the final performance depends on the MLP units; ii) **Random (R):** The average performance over 10 independent random samplings of *k head units*. This can be seen as a lower bound on the expected performance; iii) **Base (B):** The original performance of the model without any modification to its residual. Intuitively, this could represent a theoretical upper bound on the performance if there are no task-aligned units.

The greedy selection strategy scores heads independently, disregarding inter-head relationships. To make the selection aware of them, we optimize a scalar weight for each head simultaneously using gradient descent, providing an empirical upper bound for the selection performance. We refer to this procedure as **Optimized (O)**.

We evaluate these unit selection strategies on two CLIP-like models (BLIP-L and OpenCLIP-L) and 10 of the datasets of Section 3.3. We choose $k$ for the greedy and random selection so that 5% of the total heads are considered. Additional results are presented in the Appendix in Table 7, Table 4 and Table 5 for other CLIP-like models and Appendix A.3 for unimodal ones (DINOv2-L and ViT-L).

**Result analysis** As reported in Table 1, the unsupervised scoring (**U**) performs unexpectedly well despite not explicitly considering the task. This effectiveness likely stems from the fact that core task information is often embedded in the first few principal components (PCs) of the output. By aligning with this information using only a few heads, we achieve a dual benefit: preserving essential task-relevant information while effectively filtering out noise. In fact, adding the conditioning on the task (**U|T**) is just slightly beneficial in terms of performance, with the exception of SVHN. The vast majority (on average around 90%) of alignment between task and residual comes from the head contributions, as witnessed by the similarity between the columns **H** and **B**. The optimized selection strategy (**O**) is extremely powerful, having the best score among them, and sometimes almost doubling the original model performances (**B**), showing how task-relevant information is already present in the residual, just hidden.

These results illustrate that retaining only task-aligned units (effectively "panning for the gold" contained in the residual) is highly effective across the board.

### 4.2 Spectral ResiDual Alignment

Given that: i) unit specialization is essentially encoded in the principal components (Section 3.2); ii) components of general enough data distributions (e.g., ImageNet) capture specialized behavior even on other datasets (Section 3.3); iii) retaining only task-aligned units is beneficial for image-text alignment (Section 4.1), we propose a method to directly filter information along the residual at the spectral level: **ResiDual**.

In short, we start from the decomposition formula for the residual stream (Equation (1)) and allow anisotropic scaling of each unit representation. Specifically, given a unit representation $\boldsymbol{X}$, its corresponding principal component basis $\boldsymbol{\Phi}$, and associated mean $\boldsymbol{\mu}$, we define the ResiDual transformation of $\boldsymbol{X}$ as:

$$\mathrm{RD}_{\boldsymbol{\Phi},\boldsymbol{\mu}}(\boldsymbol{X}, \boldsymbol{\lambda}) = \boldsymbol{\Phi}^{-1}\mathrm{diag}(\boldsymbol{\lambda})\boldsymbol{\Phi}(\boldsymbol{X} - \boldsymbol{\mu})^{T}, \tag{6}$$

where the learnable vector $\boldsymbol{\lambda}$ contains the weights associated with each principal component. Then, the transformation is applied to every residual unit independently, resulting in a transformed version of the output $\boldsymbol{Y}$:

$$\boldsymbol{Y}' = \sum_{i=1}^{|\mathbb{U}|}\mathrm{RD}_{\boldsymbol{\Phi}_i,\boldsymbol{\mu}_i}(\boldsymbol{U}_i, \boldsymbol{\lambda}_i). \tag{7}$$

In summary, ResiDual models a simple spectral anisotropic scaling of residual units that results in more complex dynamics in the output space. In this section, we extensively evaluate the effectiveness of this method across a variety of configurations, models and datasets.

**Experimental setting** We evaluate ResiDual in 3 different configurations: i) **RD**, as presented in the original formulation; ii) **RD***, a version of ResiDual constrained on the number and type of units (we restrict it to heads) and the number of considered components (we truncate PCA bases to explain 90% of the variance). This is done because head units, and specifically their principal components, are strongly connected to specialization; iii) **RD$^{\boldsymbol{Y}}$**, where the ResiDual transformation is applied directly on the output encoding $\boldsymbol{Y}$, to assess whether output components can already be aligned with the task. For all configurations, we select ImageNet as a reference for the PCA bases $\boldsymbol{\Phi}$ and unit means $\boldsymbol{\mu}$ that appear in the ResiDual Equation (6). Please refer to Table 8 for additional experiments with dataset-specific bases.

We compare ResiDual with 3 reference alignment methods: i) **Base**: the base zero-shot performance of the model, relying on the alignment coming from pre-training; ii) **Full Finetuning**: we consider its score the empirical upper bound for alignment. It is obtained by finetuning the whole vision transformer with frozen text encodings; iii) **Linear Aligner (Lin)**: the performance of a trained linear transformation of the output shows to what extent output alignment could be linearly recovered. This approach is well-supported by recent studies, which indicate that even independently trained models with different architectures can often be aligned by a simple linear transformation (Moschella et al., 2023; Maiorca et al., 2024; Norelli et al., 2023; Lähner & Moeller, 2024; Balasubramanian et al., 2024).

For this experiment, we work with 3 CLIP-like models, BLIP-L, CLIP-L, and OpenCLIP-L (CLIP/OpenCLIP-B results can be found in the Appendix in Table 9 and Figure 21), and tune them on the 10 datasets employed in Section 4.1. All training runs use the Schedule-Free Adam optimizer (Defazio et al., 2024) with the automatic learning rate finder by Smith (2017), implemented in PyTorch Lightning (Falcon & The PyTorch Lightning team, 2019). The maximum number of epochs is 30, with an early-stopping policy on the validation set accuracy with patience of 5 epochs.

**Result analysis** A comparative analysis of ResiDual against reference alignment methods is reported in Figure 5. We observe that a linear transformation of the output is sufficient to approximate full finetuning performance. The spectral residual transformation modelled by ResiDual attains comparable (and, in some cases, better) results than the linear aligner on all datasets. These statements hold true across all models. Specifically, the case of SVHN stands out: on this input dataset, ResiDual has an advantage of approximately 10% on all models. We hypothesize that this gap is due to the absence of task-relevant features at the output level. This is confirmed by the results in Table 2: on SVHN, **RD$^{\boldsymbol{Y}}$** has the largest gap from **RD**, meaning

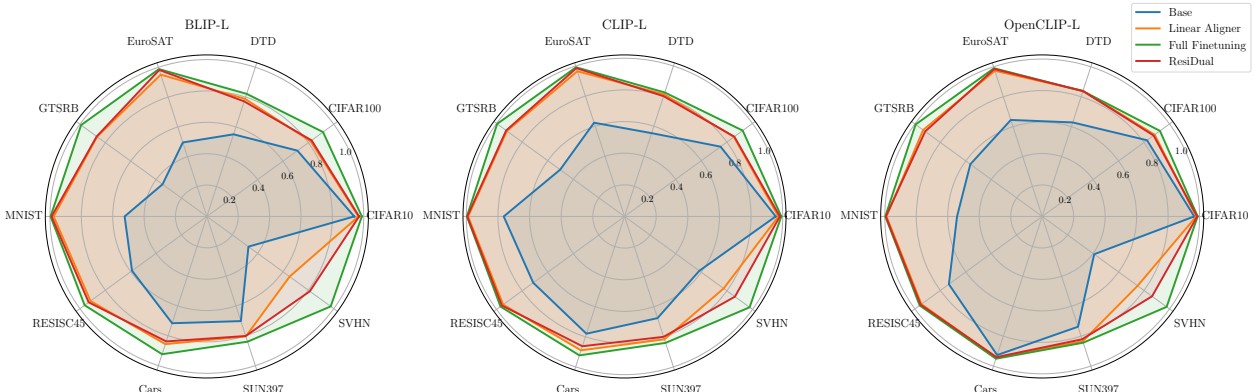

Figure 5: Performance comparison between text-image alignment methods on zero-shot classification tasks.

| Dataset | BLIP-L | | | | CLIP-L | | | | OpenCLIP-L | | | |
|---|---|---|---|---|---|---|---|---|---|---|---|---|
| | Lin | RD | RD* | RD$^Y$ | Lin | RD | RD* | RD$^Y$ | Lin | RD | RD* | RD$^Y$ |
| CIFAR10 | 0.97 | 0.97 | 0.97 | 0.96 | 0.97 | 0.98 | 0.98 | 0.97 | 0.98 | 0.98 | 0.98 | 0.98 |
| CIFAR100 | 0.82 | 0.83 | 0.81 | 0.74 | 0.85 | 0.86 | 0.85 | 0.80 | 0.88 | 0.88 | 0.87 | 0.85 |
| DTD | 0.79 | 0.77 | 0.76 | 0.58 | 0.81 | 0.80 | 0.79 | 0.63 | 0.84 | 0.84 | 0.83 | 0.69 |
| EuroSAT | 0.95 | 0.98 | 0.98 | 0.83 | 0.97 | 0.99 | 0.98 | 0.95 | 0.97 | 0.98 | 0.98 | 0.94 |
| GTSRB | 0.87 | 0.87 | 0.84 | 0.59 | 0.92 | 0.92 | 0.92 | 0.77 | 0.94 | 0.92 | 0.92 | 0.78 |
| MNIST | 0.98 | 0.99 | 0.99 | 0.93 | 0.99 | 0.99 | 0.99 | 0.97 | 0.99 | 0.99 | 0.99 | 0.97 |
| RESISC45 | 0.92 | 0.93 | 0.92 | 0.77 | 0.95 | 0.96 | 0.95 | 0.87 | 0.95 | 0.95 | 0.95 | 0.89 |
| Cars | 0.86 | 0.84 | 0.82 | 0.75 | 0.89 | 0.86 | 0.85 | 0.81 | 0.94 | 0.94 | 0.94 | 0.93 |
| SUN397 | 0.80 | 0.80 | 0.77 | 0.72 | 0.82 | 0.80 | 0.77 | 0.70 | 0.83 | 0.82 | 0.80 | 0.75 |
| SVHN | 0.65 | 0.81 | 0.77 | 0.53 | 0.77 | 0.86 | 0.86 | 0.70 | 0.75 | 0.86 | 0.85 | 0.64 |
| **Average** | 0.86 | **0.88** | 0.86 | 0.74 | 0.89 | **0.90** | 0.90 | 0.82 | 0.91 | **0.92** | 0.91 | 0.84 |
| **#params** | 65.8k | 30.7k | 8.3k | 256 | 590k | 43k | 14k | 768 | 590k | 43k | 13.2k | 768 |

Table 2: Accuracy produced by different configurations of ResiDual, compared with a linear aligner. (Lin): linear aligner at the output level; (RD): ResiDual in the original formulation (all principal components of all residual units); (RD*): ResiDual limited to head units alone (no MLPs), and PCs truncated to 90% of explained variance; (RD$^Y$): ResiDual applied to the output encoding.

that output components are not well aligned with the task. While having approximately 30% of the learnable parameters, **RD*** achieves comparable results to **RD**, indicating that applying the ResiDual procedure to heads (and not MLPs) and their first principal components alone is enough. Moreover, we note that in the cases where the performance of the original model is already satisfactory (e.g., CIFAR-10), applying ResiDual does not compromise alignment.

Overall, these results show that, by leveraging spectral-level operations along the residual, ResiDual builds a concise yet expressive transformation that bridges the modality gap, closely approximating full finetuning-level performance, even in its more parameter-efficient configuration.

# 5 Conclusions

In this work, we analyzed the emergent specialization property of attention heads in vision transformers and unveiled its connection with the spectral geometry of residual representations. Specifically, we focused on the relationship between head specialization and downstream task performance. Then, we leveraged this to introduce ResiDual, a method that we employed to improve alignment in multimodal transformers by applying

spectral anisotropic scaling along the residual stream. ResiDual proved effective in emphasizing task-relevant principal components and dampening down the others, akin to panning for gold in the residual stream.

**Limitations** ResiDual fundamentally works by extracting information from residual units already containing task-relevant principal components. We expect that when this assumption does not hold, ResiDual cannot recover the alignment, resulting in a significant drop in downstream performance. Moreover, our downstream tasks focused on zero-shot image classification. This implies considering only the alignment between `[CLS]` token and classifier, ignoring sequence-level information. However, preliminary results show high spectral compatibility between the `[CLS]` representations and the other patch tokens (Figure 12).

**Future work** The ResiDual formulation is based solely on the residual decomposition technique, which opens to its application across virtually any transformer architecture (with some preliminary results on unimodal encoders in Appendix A.3 ). Our findings show that ResiDual can be constrained at the coarse level to operate on a subset of units (i.e., attention heads) and at the fine level on specific principal components. Additional constraints to focus on the first few model layers create opportunities to enhance model inference, while the possibility to disable specific spectral components opens up fairness applications.

**Acknowledgements** The authors gratefully acknowledge Volkan Cevher for an insightful discussion about sparse recovery algorithms, Alex Smola for valuable feedback on the experiments, and Marco Baroni for an engaging conversation on the phenomenon of head specialization in NLP.

Luca Bortolussi was supported by the PNRR project iNEST (Interconnected Nord-Est Innovation Ecosystem) funded by the European Union NextGenerationEU (Piano Nazionale di Ripresa e Resilienza (PNRR) – Missione 4 Componente 2, Investimento 1.5 – D.D. 1058 23/06/2022, ECS 00000043, CUP J43C22000320006).

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

# A    Appendix

## A.1    Residual decomposition

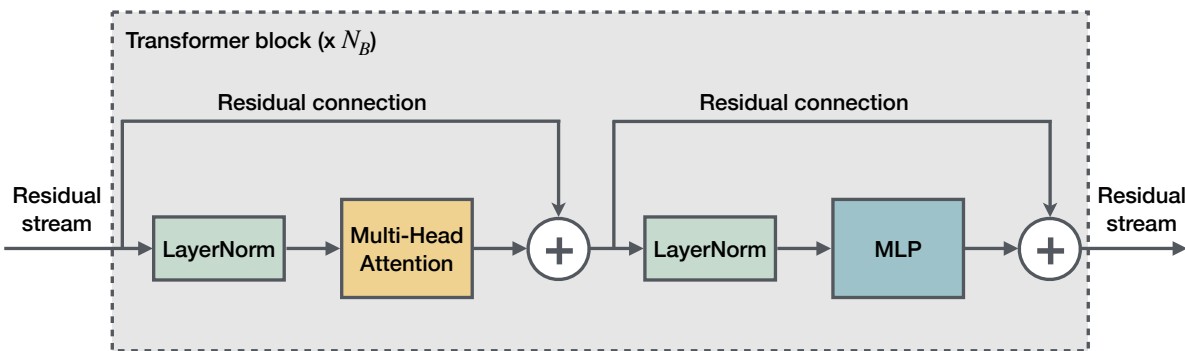

Figure 6: Overview of a Transformer block. Multi-Head Attention and MLP are both surrounded by residual connections and LayerNorm is applied before each sub-layer.

Transformers are residual networks made of stacked blocks that contain a Multi-Head Attention (MHA) layer and an MLP, both surrounded by a residual connection. All vision transformers we employ in this work abide by the "pre-norm" Wang et al. (2019) architecture, which slightly modified the original (Vaswani et al., 2017) by moving LayerNorm before MHA and MLP sub-layers (Figure 6). This modification implies that, at each layer, MHA and MLP sub-layers directly write their output representation to the residual stream. Hence, the final latent representation of the model is given by:

$$\boldsymbol{Y} = \boldsymbol{X}_0 + \sum_{i=1}^{L} \boldsymbol{A}_i + \sum_{i=1}^{L} \boldsymbol{M}_i$$

where $\boldsymbol{X}_0$ is the initial data embedding, $\boldsymbol{A}_i$ is the attention output of layer $i$ and $\boldsymbol{M}_i$ is the MLP output at layer $i$. All these encodings share the same dimensionality $d$ with the model output.

Attention representations can be further decomposed into head contributions, as they are the result of linear operations applied to the head-level representations. Specifically, each attention head $h$ of $N_h$ at layer $i$ produces a representation $\boldsymbol{H}_{i,h} = \mathrm{Softmax}\left(\frac{\boldsymbol{Q}_{i,h}\boldsymbol{K}_{i,h}^T}{\sqrt{d_k}}\right)\boldsymbol{V}_{i,h}$, whose dimension is $\frac{d}{N_h}$. Contributions for all heads are then concatenated and projected linearly to obtain the MHA output:

$$\boldsymbol{A}_i = (\mathrm{cat}(\boldsymbol{H}_{i,1}, ..., \boldsymbol{H}_{i,N_h}))\boldsymbol{W}_i + \boldsymbol{b}_i$$

Assuming that the bias term $\boldsymbol{b}$ can be split equally between heads, this operation can be equivalently expressed in a distributed form:

$$\boldsymbol{A}_i = \sum_{h=1}^{N_h} \hat{\boldsymbol{H}}_{i,h} = \sum_{h=1}^{N_h} (\boldsymbol{H}_{i,h}^0 \boldsymbol{W}_i + \frac{\boldsymbol{b}_i}{N_h})$$

These terms $\hat{\boldsymbol{H}}_{i,j}$ are the *head representations* we consider in this paper. They have the same dimensionality $d$ as the residual stream (and model output), and they are simply obtained by linearly projecting the 'raw'

head contributions, properly padded with 0s to match the dimensionality of the residual stream. Hence, we arrive at this final decomposition:

$$\boldsymbol{Y} = \boldsymbol{X}_0 + \sum_{i=1}^{L}\sum_{h=1}^{N_h} \hat{\boldsymbol{H}}_{i,h} + \sum_{i=1}^{L} \boldsymbol{M}_i$$

In many transformer models, such as the ones employed in this work, the final residual encoding $\boldsymbol{Y}$ is not the final output of the model. For instance, in CLIP, $\boldsymbol{Y}$ is passed through a LayerNorm and then through a bias-less linear projection $\boldsymbol{P}$ that maps ViT encodings to the shared vision-language space:

$$\hat{\boldsymbol{Y}} = \boldsymbol{P}(\text{LayerNorm}(\boldsymbol{Y}))$$

However, this operation can be again distributed over the summands that produce $\boldsymbol{Y}$ because of the linearity of the final projection and because LayerNorm can be rewritten as an affine transformation, as in Gandelsman et al. (2024). The representations we employ in this paper are mapped to the output space by applying (if present) the final projection and LayerNorm to each unit entry ($\hat{\boldsymbol{H}}_{i,h}$ or $\boldsymbol{M}_i$).

## A.2 Connecting TextSpan and Matching Pursuit

---

**Algorithm 1:** TextSpan (Gandelsman et al., 2024)

---

**Input :** Signal Matrix $\boldsymbol{X} \in \mathbb{R}^{n,d}$, dictionary $\boldsymbol{D} \in \mathbb{R}^{k,d}$, number of iterations $N$.
**Output:** Reconstruction $\boldsymbol{X}_r^N$, support set $\mathbb{C}^N$
**Initialization:** Residual $\boldsymbol{R}^0 = \boldsymbol{X}$, reconstruction $\boldsymbol{X}_r^0 = \boldsymbol{0}$, dictionary $\boldsymbol{D}^0 = \boldsymbol{D}$, support set $\mathbb{C}^0 = \emptyset$ ;
**for** $t \in \{0, ..., N-1\}$ **do**
$\quad$ $\boldsymbol{P} \leftarrow \boldsymbol{D}^t \boldsymbol{R}^{t^T}$;
$\quad$ $p^t \leftarrow \arg\max_{j=1}^{k} \text{Var}(\boldsymbol{P}[j])$;
$\quad$ $\mathbb{C}^{t+1} \leftarrow \mathbb{C}^t \cup \{p^t\}$;
$\quad$ $\boldsymbol{R}^{t+1} \leftarrow \boldsymbol{R}^t - \text{proj}(\boldsymbol{R}^t, \boldsymbol{D}^t[p^t])$;
$\quad$ $\boldsymbol{X}_r^{t+1} \leftarrow \boldsymbol{X}_r^t + \text{proj}(\boldsymbol{R}^t, \boldsymbol{D}^t[p^t])$;
$\quad$ $\boldsymbol{D}^{t+1} \leftarrow \boldsymbol{D}^t - \text{proj}(\boldsymbol{D}^t, \boldsymbol{D}^t[p^t])$;
**end**

---

**Algorithm 2:** Simultaneous Orthogonal Matching Pursuit (SOMP) (Tropp et al., 2006)

---

**Input :** Signal Matrix $\boldsymbol{X} \in \mathbb{R}^{n,d}$, dictionary $\boldsymbol{D} \in \mathbb{R}^{k,d}$, number of iterations $N$.
**Output:** Reconstruction $\boldsymbol{X}_r^N$, support set $\mathbb{C}^N$
**Initialization:** Residual $\boldsymbol{R}^0 = \boldsymbol{X}$, reconstruction $\boldsymbol{X}_r^0 = \boldsymbol{0}$, support set $\mathbb{C}^0 = \emptyset$;
**for** $t \in \{0, ..., N-1\}$ **do**
$\quad$ $\boldsymbol{P} \leftarrow \boldsymbol{D} \boldsymbol{R}^{t^T}$;
$\quad$ $p^t \leftarrow \arg\max_{j=1}^{k}(||\boldsymbol{P}[j]||_1)$;
$\quad$ $\mathbb{C}^{t+1} \leftarrow \mathbb{C}^t \cup \{p^t\}$;
$\quad$ $\boldsymbol{W}^t \leftarrow \arg\min_{\boldsymbol{W}} ||\boldsymbol{X} - \boldsymbol{W}\boldsymbol{D}[\mathbb{C}^t]||_F$;
$\quad$ $\boldsymbol{X}_r^{t+1} \leftarrow \boldsymbol{W}^t \boldsymbol{D}[\mathbb{C}^t]$;
$\quad$ $\boldsymbol{R}^{t+1} \leftarrow \boldsymbol{X} - \boldsymbol{X}_r^{t+1}$;
**end**

---

The TextSpan algorithm was introduced in Gandelsman et al. (2024) to find a decomposition of CLIP heads on a set of textual descriptions. Here, we show that TextSpan is equivalent to Simultaneous Orthogonal Matching Pursuit (SOMP) (Tropp et al., 2006), with a few light modifications.

The first modification is that, in TextSpan, before computing the decomposition, the dictionary is filtered through a projection on the first principal components of the signal. Output-level text encodings have high

semantic granularity: this operation results in a dictionary restricted to the head span. In the following, we will consider this a dictionary preprocessing step and assume that SOMP and TextSpan are provided with the same dictionary, filtered or not.

The second modification is that in TextSpan, the row variance of $\boldsymbol{D}\boldsymbol{R}^T$ is used instead of the $\ell_1$ norm as a criterion for atom selection. Here, we show that the two algorithms are equivalent if this criterion is applied in SOMP (or vice versa, the $\ell_1$ in TextSpan).

We are given a signal matrix $\boldsymbol{X} \in \mathbb{R}^{n,d}$ and two (initially identical) dictionaries $\boldsymbol{D}_{MP} = \boldsymbol{D}_{TS}^0 \in \mathbb{R}^{k,d}$.

We will proceed by induction. At $t = 0$, the two methods pick the same atom $p_0$ (which enters the support set $\mathbb{C}^1$), and identically update the residual:

$$\mathbb{C}_{MP}^1 = \{p_0\}, \quad \boldsymbol{R}_{MP}^1 = \boldsymbol{X} - \text{proj}(\boldsymbol{X}, \boldsymbol{D}_{MP}[p_0])$$

$$\mathbb{C}_{TS}^1 = \{p_0\}, \quad \boldsymbol{R}_{TS}^1 = \boldsymbol{X} - \text{proj}(\boldsymbol{X}, \boldsymbol{D}_{TS}^0[p_0]) = \boldsymbol{R}_{MP}^1, \quad \boldsymbol{D}_{TS}^1 \perp \boldsymbol{D}_{TS}^0[p_0]$$

Now, suppose we are at step $t = n$ with $\mathbb{C}_{MP}^n = \mathbb{C}_{TS}^n$ and $\boldsymbol{R}_{TS}^n = \boldsymbol{R}_{MP}^n$.

The two dictionaries will be different: $\boldsymbol{D}_{MP}$ never gets updated, while $\boldsymbol{D}_{TS}^n \perp \boldsymbol{D}_{TS}^0[\mathbb{C}_{TS}^n]$ because TextSpan applies a Gram–Schmidt process that finds an orthogonal basis for the subspace of selected atoms. Since in TextSpan the dictionary is orthogonalized at each step, at time $n$ it is orthogonal to *all* previously chosen atoms (which are also orthogonal to each other). The dictionary of SOMP can be decomposed into two terms, one contained in the span of atoms chosen until this point by TextSpan and one orthogonal, which corresponds to the current dictionary of TextSpan:

$$\boldsymbol{D}_{MP} = \boldsymbol{D}_{MP,\|} + \boldsymbol{D}_{MP,\perp} = \boldsymbol{D}_{MP,\|} + \boldsymbol{D}_{TS}^n$$

The residual of TextSpan (which, by inductive hypothesis, is identical to the residual of SOMP) is, by definition, orthogonal to the atoms already chosen by TextSpan. Hence the selection step of SOMP will compute:

$$\boldsymbol{D}_{MP}\boldsymbol{R}_{MP}^n{}^T = (\boldsymbol{D}_{MP,\|} + \boldsymbol{D}_{MP,\perp})\boldsymbol{R}_{MP}^n{}^T = \boldsymbol{D}_{MP,\perp}\boldsymbol{R}_{TS}^n{}^T = \boldsymbol{D}_{TS}^n\boldsymbol{R}_{TS}^n{}^T$$

Then, at step $n+1$, the two algorithms will pick the same atom index again and update identically the residual, removing its projection on the chosen atom, i.e., $\mathbb{C}_{MP}^{n+1} = \mathbb{C}_{TS}^{n+1}$. The last thing to prove is that the residual is also the same. For SOMP, the least squares solution of the optimization problem results in:

$$\boldsymbol{R}_{MP}^{n+1} = \boldsymbol{X} - \boldsymbol{X}\boldsymbol{D}_{MP}^T[\mathbb{C}_{MP}^{n+1}](\boldsymbol{D}_{MP}[\mathbb{C}_{MP}^{n+1}]\boldsymbol{D}_{MP}^T[\mathbb{C}_{MP}^{n+1}])^{-1}\boldsymbol{D}_{MP}[\mathbb{C}_{MP}^{n+1}] = \boldsymbol{X} - \text{proj}(\boldsymbol{X}, \boldsymbol{D}_{MP}[\mathbb{C}_{MP}^{n+1}])$$

While for TextSpan, we get:

$$\boldsymbol{R}_{TS}^{n+1} = \boldsymbol{R}_{TS}^n - \text{proj}(\boldsymbol{R}_{TS}^n, \boldsymbol{D}_{TS}^n[p_{n+1}]) = \boldsymbol{R}_{TS}^n - \text{proj}(\boldsymbol{R}_{TS}^n, \boldsymbol{D}_{TS}^n[\mathbb{C}_{TS}^{n+1}]) = \boldsymbol{R}_{TS}^n - \text{proj}(\boldsymbol{R}_{TS}^n, \boldsymbol{D}_{TS}^0[\mathbb{C}_{TS}^{n+1}])$$

Where the first equality is the definition of the residual; the second is because $\boldsymbol{D}_{TS}^n[p_{n+1}] \perp \boldsymbol{D}_{TS}^n[\mathbb{C}_{TS}^n]$; the last is because the residual and the last chosen atom $\boldsymbol{D}_{TS}^n[p_{n+1}]$ are orthogonal to $\boldsymbol{D}_{TS}^0[\mathbb{C}_{TS}^n]$. Now, we can use the inductive hypothesis $\boldsymbol{R}_{TS}^n = \boldsymbol{R}_{MP}^n$, that $\mathbb{C}_{TS}^{n+1} = \mathbb{C}_{MP}^{n+1}$ and $\boldsymbol{D}_{TS}^0 = \boldsymbol{D}_{MP}$, so:

$$\begin{aligned}
\boldsymbol{R}_{TS}^{n+1} &= \boldsymbol{R}_{MP}^n - \text{proj}(\boldsymbol{R}_{MP}^n, \boldsymbol{D}_{MP}[\mathbb{C}_{MP}^{n+1}]) \\
&\overset{\text{MP residual}}{=} \boldsymbol{X} - \text{proj}(\boldsymbol{X}, \boldsymbol{D}_{MP}[\mathbb{C}_{MP}^n]) - \text{proj}(\boldsymbol{X} - \text{proj}(\boldsymbol{X}, \boldsymbol{D}_{MP}[\mathbb{C}_{MP}^n]), \boldsymbol{D}_{MP}[\mathbb{C}_{MP}^{n+1}]) \\
&\overset{\text{proj is linear}}{=} \boldsymbol{X} - \text{proj}(\boldsymbol{X}, \boldsymbol{D}_{MP}[\mathbb{C}_{MP}^n]) - \text{proj}(\boldsymbol{X}, \boldsymbol{D}_{MP}[\mathbb{C}_{MP}^{n+1}]) + \text{proj}(\text{proj}(\boldsymbol{X}, \boldsymbol{D}_{MP}[\mathbb{C}_{MP}^n]), \boldsymbol{D}_{MP}[\mathbb{C}_{MP}^{n+1}]) \\
&\overset{\text{MP residual}}{=} \boldsymbol{R}_{MP}^{n+1} - \text{proj}(\boldsymbol{X}, \boldsymbol{D}_{MP}[\mathbb{C}_{MP}^n]) + \text{proj}(\text{proj}(\boldsymbol{X}, \boldsymbol{D}_{MP}[\mathbb{C}_{MP}^n]), \boldsymbol{D}_{MP}[\mathbb{C}_{MP}^{n+1}]) \\
&= \boldsymbol{R}_{MP}^{n+1} - \text{proj}(\boldsymbol{X}, \boldsymbol{D}_{MP}[\mathbb{C}_{MP}^n]) + \text{proj}(\boldsymbol{X}, \boldsymbol{D}_{MP}[\mathbb{C}_{MP}^n])]) \\
&= \boldsymbol{R}_{MP}^{n+1}
\end{aligned}$$

where the second to last equality comes from the fact that $\text{proj}(\boldsymbol{X}, \boldsymbol{D}_{MP}[\mathbb{C}_{MP}^n])$ is already in a subspace of $\boldsymbol{D}_{MP}[\mathbb{C}_{MP}^{n+1}]$, so the outer projection does not change the result of the inner one.

## A.3 Prototypical Alignment

In this section, we provide results in a different experimental setting, designed for applying our framework to unimodal encoders such as DINOv2 and ViT.

For these encoders, instead of deriving task boundaries from text encodings, we use a prototypical approach (Snell et al., 2017): for each dataset, we average the latent representations of all the corresponding samples in the training set for each target class. This produces representations that contain the average and high-level information for that class, similar to a textual encoding of a simple caption, "An image of <class>".

| | DINOv2-L | | | | | | | ViT-L | | | | | | |
|---|---|---|---|---|---|---|---|---|---|---|---|---|---|---|
| **Dataset** | U | U\|T | S | R | H | B | O | U | U\|T | S | R | H | B | O |
| CIFAR10 | 0.85 | 0.92 | 0.95 | 0.51 | 0.95 | 0.96 | 0.99 | 0.93 | 0.93 | 0.91 | 0.62 | 0.94 | 0.96 | 0.97 |
| CIFAR100 | 0.77 | 0.79 | 0.78 | 0.25 | 0.84 | 0.88 | 0.89 | 0.68 | 0.70 | 0.70 | 0.34 | 0.81 | 0.81 | 0.86 |
| Cars | 0.43 | 0.43 | 0.44 | 0.05 | 0.46 | 0.43 | 0.72 | 0.24 | 0.26 | 0.27 | 0.07 | 0.36 | 0.28 | 0.48 |
| DTD | 0.65 | 0.65 | 0.65 | 0.30 | 0.67 | 0.70 | 0.76 | 0.50 | 0.50 | 0.51 | 0.28 | 0.59 | 0.58 | 0.67 |
| EuroSAT | 0.71 | 0.74 | 0.71 | 0.36 | 0.75 | 0.71 | 0.95 | 0.65 | 0.64 | 0.71 | 0.54 | 0.73 | 0.68 | 0.94 |
| GTSRB | 0.21 | 0.21 | 0.36 | 0.10 | 0.27 | 0.29 | 0.66 | 0.13 | 0.16 | 0.34 | 0.14 | 0.28 | 0.20 | 0.69 |
| MNIST | 0.62 | 0.62 | 0.55 | 0.24 | 0.68 | 0.65 | 0.93 | 0.72 | 0.71 | 0.77 | 0.51 | 0.76 | 0.75 | 0.94 |
| RESISC45 | 0.76 | 0.75 | 0.76 | 0.25 | 0.78 | 0.74 | 0.87 | 0.50 | 0.47 | 0.50 | 0.29 | 0.68 | 0.57 | 0.79 |
| SUN397 | 0.60 | 0.60 | 0.60 | 0.15 | 0.63 | 0.68 | 0.72 | 0.48 | 0.51 | 0.51 | 0.19 | 0.67 | 0.68 | 0.71 |
| SVHN | 0.19 | 0.20 | 0.27 | 0.14 | 0.19 | 0.23 | 0.48 | 0.28 | 0.31 | 0.38 | 0.23 | 0.33 | 0.29 | 0.54 |
| **Average** | 0.58 | 0.59 | **0.61** | 0.23 | 0.62 | 0.63 | 0.80 | 0.51 | 0.52 | **0.56** | 0.32 | 0.62 | 0.58 | 0.76 |

Table 3: Accuracy when doing **zero ablation of all units except top 5%** of attention heads. Heads are assigned a binary weight using an Unsupervised (U), Task-conditioned Unsupervised (U|T), Supervised (S), and Random (R) strategy (a mean over 10 different seeds). H corresponds to using all the attention heads available, B is the original model performance, and O is the optimized continuous weighting case.

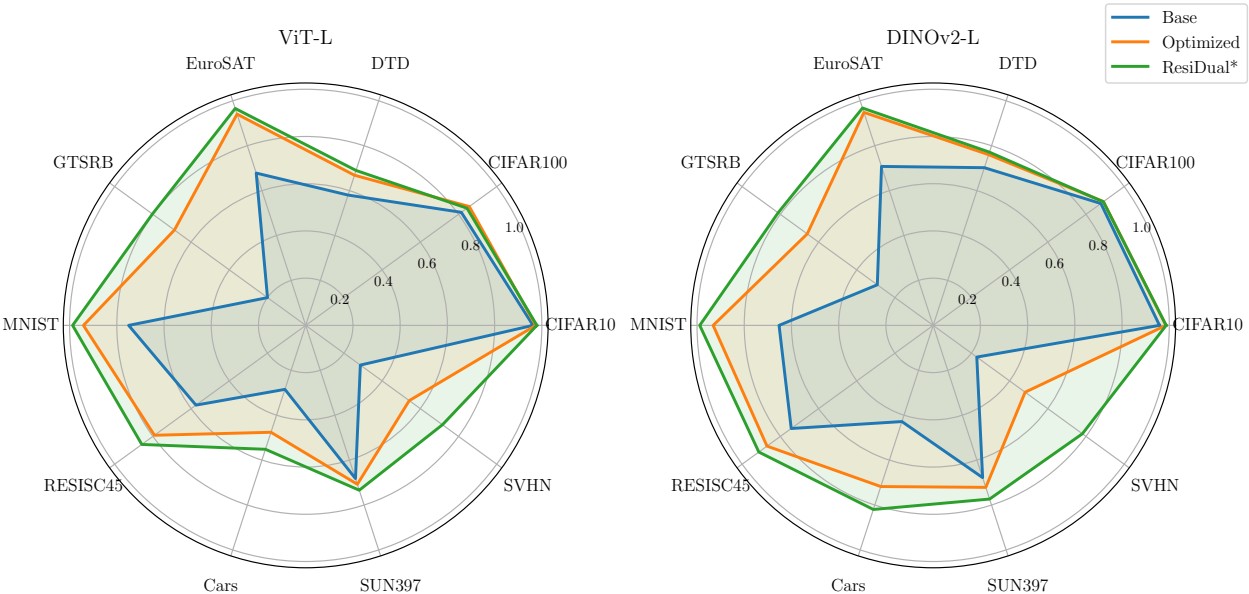

Figure 7: Performance comparison between image-prototype alignment methods on zero-shot classification tasks. **Base**: original model. **Optimized**: learned unit-wise weights (as in Section 4.1). **ResiDual\***: restricted ResiDual method as described in Section 4.2

## A.4 Additional results

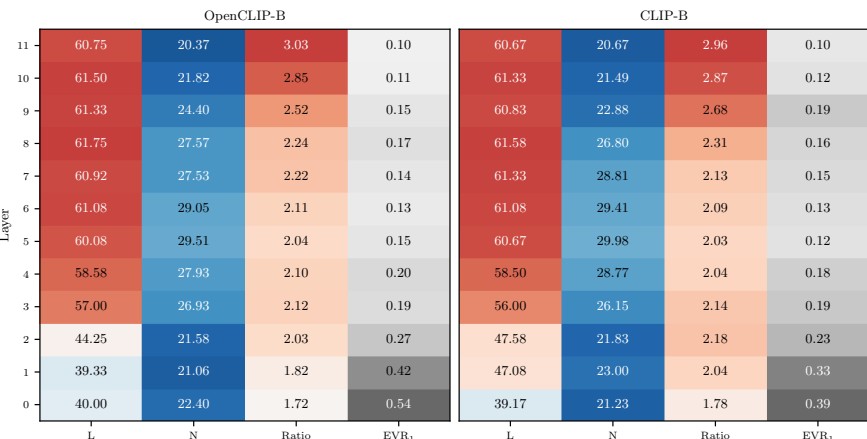

Figure 8: Heads in early layers show low-dimensional, linear structures, as suggested by similar intrinsic dimension estimates from PCA (L) and TwoNN (N). Moving toward the output layer, the true dimensionality peaks and then decreases, while PCA's linear estimate continues to rise, indicating increasing nonlinearity in head manifolds (Ratio = $\frac{L}{N}$). The first principal component ($EVR_1$) explains around 50% of the variance in early layers, dropping to around 10% in later layers.

| | TextSpan | OMP$_2$ |
|---|---|---|
| | **L20.H8** ("Scenery") | **L20.H8** ($Z = 0.35$) |
| | Photo taken in Galápagos Islands | Picture taken in the Swiss chocolate factories |
| | Image taken in Norway | Image with Mayan-inspired designs |
| | Evocative beauty | Stark and minimalist urban scene |
| | Vibrant urban energy | serene oceanside scene |
| | A skirt | Vivid cultural ceremony |
| | **L21.H11** ("Location") | **L21.H11** ($Z = 1.47$) |
| | Picture taken in Cyprus | Picture taken in Hungary |
| | Picture taken in Ontario, Canada | Photo taken in the Californian vineyards |
| | Photo taken in Rio de Janeiro, Brazil | serene woodland refuge |
| | Photo captured in the Arizona desert | Photo taken in the Australian rainforest |
| | Picture captured in the Scottish highlands | Photo taken in Canadian Rockies |
| | **L22.H8** ("Letters") | **L22.H8** ($Z = 1.84$) |
| | A photo with the letter F | A photo with the letter G |
| | A photo with the letter V | A photo with the letter J |
| | A photo with the letter D | Photo taken in Monument Valley |
| | A photo with the letter T | Enchanting fantasy world |
| | A photo with the letter X | A labyrinth |
| | **L23.H2** ("Animals") | **L23.H2** ($Z = 1.11$) |
| | Image showing prairie grouse | A capacitor |
| | Image with a penguin | A spiky texture |
| | A magnolia | A wolf |
| | An image with dogs | Image with an ant |
| | An image with cats | A spirograph-like shape |

Figure 9: Comparison between TextSpan and Orthogonal Matching Pursuit on the second principal component (OMP$_2$), applied to the heads of OpenCLIP-L. *Left*: agreement score between the descriptions returned by the two methods. *Right*: qualitative comparison of selected descriptions for 4 heads, one per layer, at different agreement levels.

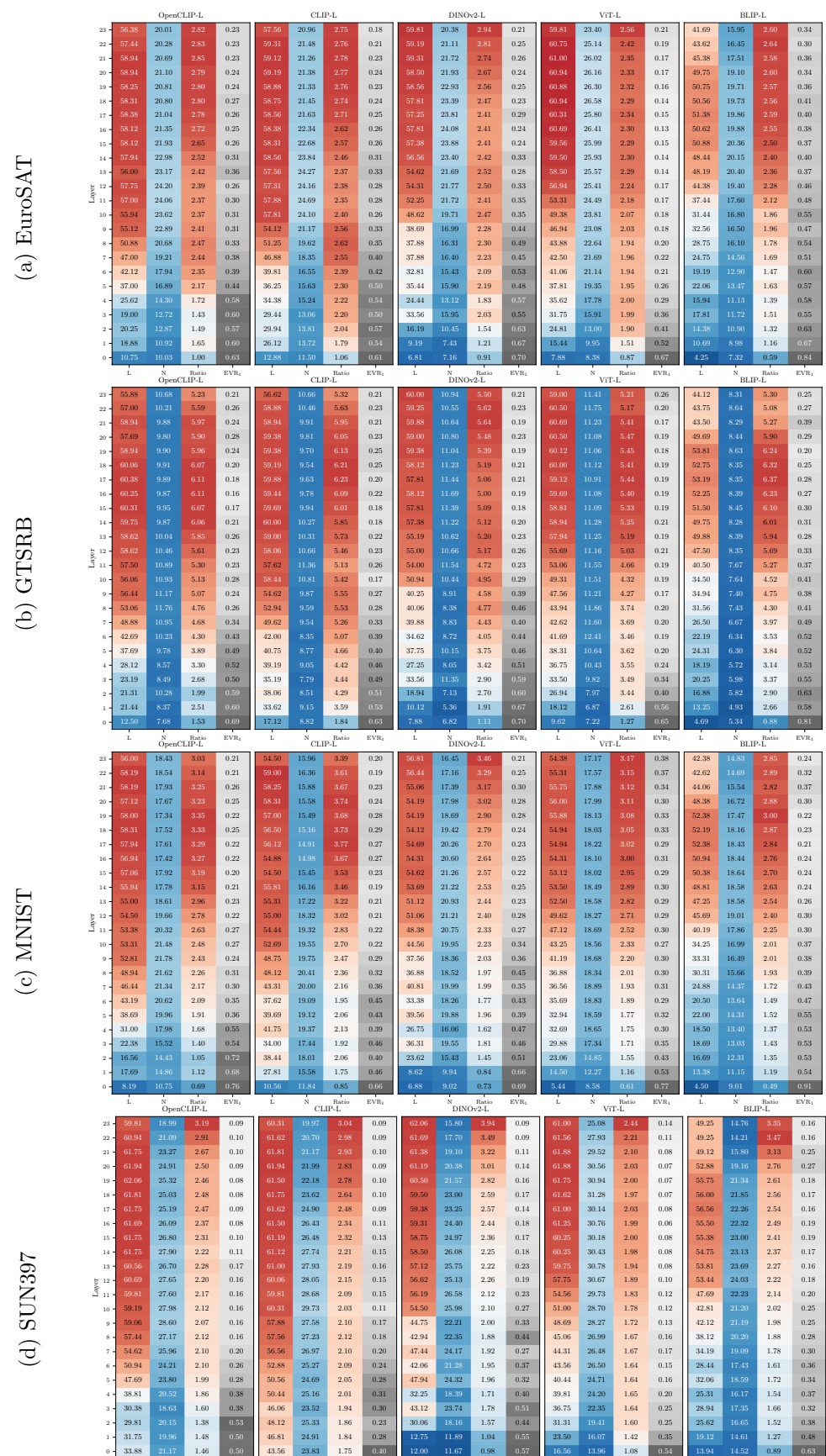

Figure 10: Additional datasets for the study on the residual dimensionality described in Section 3.1

| Layer | OpenCLIP-L | | | | CLIP-L | | | | DINOv2-L | | | | ViT-L | | | | BLIP-L | | | |
|---|---|---|---|---|---|---|---|---|---|---|---|---|---|---|---|---|---|---|---|---|
| | L | N | Ratio | $EVR_1$ | L | N | Ratio | $EVR_1$ | L | N | Ratio | $EVR_1$ | L | N | Ratio | $EVR_1$ | L | N | Ratio | $EVR_1$ |
| 23 | 474.00 | 17.59 | 26.95 | 0.12 | 533.00 | 19.54 | 27.28 | 0.10 | 917.00 | 15.37 | 59.67 | 0.03 | 922.00 | 19.11 | 48.25 | 0.05 | 184.00 | 15.98 | 11.52 | 0.09 |
| 22 | 494.00 | 23.57 | 20.96 | 0.05 | 565.00 | 24.13 | 23.41 | 0.05 | 853.00 | 15.04 | 56.73 | 0.02 | 918.00 | 23.26 | 39.47 | 0.01 | 205.00 | 16.20 | 12.65 | 0.23 |
| 21 | 471.00 | 13.26 | 35.53 | 0.12 | 571.00 | 27.29 | 20.92 | 0.04 | 847.00 | 20.09 | 42.15 | 0.09 | 931.00 | 26.87 | 34.65 | 0.01 | 61.00 | 15.24 | 4.00 | 0.55 |
| 20 | 458.00 | 11.80 | 38.80 | 0.19 | 569.00 | 27.86 | 20.42 | 0.04 | 800.00 | 21.16 | 37.81 | 0.06 | 934.00 | 29.06 | 32.14 | 0.01 | 64.00 | 17.05 | 3.75 | 0.32 |
| 19 | 478.00 | 16.27 | 29.39 | 0.04 | 568.00 | 30.90 | 18.38 | 0.04 | 713.00 | 19.84 | 35.93 | 0.07 | 939.00 | 32.62 | 28.79 | 0.01 | 41.00 | 18.87 | 2.17 | 0.80 |
| 18 | 493.00 | 29.96 | 16.45 | 0.09 | 557.00 | 34.63 | 16.09 | 0.05 | 691.00 | 24.20 | 28.56 | 0.06 | 941.00 | 36.27 | 25.94 | 0.01 | 123.00 | 23.97 | 5.13 | 0.61 |
| 17 | 457.00 | 26.09 | 17.52 | 0.27 | 545.00 | 35.61 | 15.31 | 0.05 | 554.00 | 24.80 | 22.34 | 0.10 | 940.00 | 41.08 | 22.88 | 0.01 | 191.00 | 26.37 | 7.24 | 0.20 |
| 16 | 428.00 | 21.02 | 20.36 | 0.17 | 533.00 | 35.45 | 15.03 | 0.06 | 411.00 | 25.30 | 16.24 | 0.10 | 939.00 | 42.94 | 21.87 | 0.01 | 199.00 | 27.14 | 7.33 | 0.10 |
| 15 | 426.00 | 20.65 | 20.63 | 0.07 | 525.00 | 35.97 | 14.59 | 0.08 | 303.00 | 24.77 | 12.23 | 0.11 | 934.00 | 44.44 | 21.02 | 0.01 | 194.00 | 26.77 | 7.25 | 0.16 |
| 14 | 465.00 | 24.67 | 18.85 | 0.10 | 512.00 | 34.67 | 14.77 | 0.09 | 192.00 | 24.83 | 7.73 | 0.17 | 925.00 | 45.35 | 20.40 | 0.01 | 181.00 | 26.05 | 6.95 | 0.19 |
| 13 | 458.00 | 25.84 | 17.72 | 0.12 | 516.00 | 34.61 | 14.91 | 0.09 | 160.00 | 24.15 | 6.63 | 0.28 | 898.00 | 40.50 | 22.18 | 0.01 | 164.00 | 24.93 | 6.58 | 0.23 |
| 12 | 445.00 | 27.33 | 16.29 | 0.12 | 492.00 | 33.54 | 14.67 | 0.12 | 113.00 | 22.53 | 5.02 | 0.44 | 866.00 | 37.28 | 23.23 | 0.01 | 159.00 | 24.20 | 6.57 | 0.18 |
| 11 | 439.00 | 29.60 | 14.83 | 0.13 | 466.00 | 32.59 | 14.30 | 0.13 | 183.00 | 25.34 | 7.22 | 0.30 | 786.00 | 32.60 | 24.11 | 0.02 | 138.00 | 23.04 | 5.99 | 0.29 |
| 10 | 415.00 | 30.84 | 13.46 | 0.09 | 425.00 | 34.81 | 12.21 | 0.15 | 116.00 | 22.70 | 5.11 | 0.27 | 704.00 | 31.28 | 22.50 | 0.04 | 102.00 | 21.70 | 4.70 | 0.41 |
| 9 | 373.00 | 31.25 | 11.94 | 0.18 | 343.00 | 33.44 | 10.26 | 0.28 | 49.00 | 20.12 | 2.44 | 0.76 | 641.00 | 30.19 | 21.24 | 0.03 | 89.00 | 20.71 | 4.30 | 0.35 |
| 8 | 332.00 | 30.45 | 10.90 | 0.19 | 306.00 | 31.29 | 9.78 | 0.18 | 32.00 | 18.14 | 1.76 | 0.80 | 557.00 | 28.64 | 19.45 | 0.05 | 91.00 | 21.39 | 4.25 | 0.38 |
| 7 | 290.00 | 29.76 | 9.75 | 0.16 | 279.00 | 31.83 | 8.76 | 0.25 | 15.00 | 17.58 | 0.85 | 0.84 | 494.00 | 27.30 | 18.10 | 0.07 | 103.00 | 21.78 | 4.73 | 0.24 |
| 6 | 219.00 | 27.54 | 7.95 | 0.23 | 215.00 | 28.73 | 7.48 | 0.31 | 35.00 | 15.79 | 2.22 | 0.54 | 384.00 | 25.66 | 14.97 | 0.08 | 72.00 | 19.59 | 3.68 | 0.26 |
| 5 | 159.00 | 24.96 | 6.37 | 0.27 | 125.00 | 25.86 | 4.83 | 0.61 | 32.00 | 16.06 | 1.99 | 0.66 | 280.00 | 25.57 | 10.95 | 0.11 | 80.00 | 19.45 | 4.11 | 0.17 |
| 4 | 67.00 | 17.69 | 3.79 | 0.43 | 155.00 | 26.48 | 5.85 | 0.33 | 33.00 | 14.74 | 2.24 | 0.30 | 193.00 | 23.90 | 8.07 | 0.18 | 53.00 | 16.68 | 3.18 | 0.25 |
| 3 | 39.00 | 16.56 | 2.36 | 0.36 | 131.00 | 21.70 | 6.04 | 0.32 | 16.00 | 13.39 | 1.19 | 0.63 | 124.00 | 22.31 | 5.56 | 0.25 | 42.00 | 16.20 | 2.59 | 0.27 |
| 2 | 56.00 | 27.15 | 2.06 | 0.85 | 42.00 | 13.12 | 3.20 | 0.30 | 9.00 | 10.55 | 0.85 | 0.76 | 89.00 | 18.70 | 4.76 | 0.16 | 52.00 | 19.23 | 2.70 | 0.31 |
| 1 | 15.00 | 14.89 | 1.01 | 0.74 | 37.00 | 14.31 | 2.59 | 0.40 | 7.00 | 9.16 | 0.76 | 0.85 | 56.00 | 15.78 | 3.55 | 0.25 | 40.00 | 17.09 | 2.34 | 0.38 |
| 0 | 6.00 | 9.03 | 0.66 | 0.81 | 8.00 | 12.56 | 0.64 | 0.80 | 12.00 | 10.27 | 1.17 | 0.65 | 31.00 | 13.04 | 2.38 | 0.22 | 27.00 | 15.35 | 1.76 | 0.39 |

(a) MLP dimensionality visualization for OpenCLIP-L on ImageNet.

| L20.MLP | L21.MLP |
|---|---|
| Picture taken in a zoo or wildlife sanctuary | Serene meadow landscape |
| Intense motorsport race | Evocative streets |
| Nostalgic vibe | A shelf |
| Balanced composition | Image captured in the Patagonian wilderness |
| Intricate architectural carving | Graceful swimming fish |

| L22.MLP | L23.MLP |
|---|---|
| Photo taken in Bangkok, Thailand | Image showing prairie grouse |
| Serene wilderness | A photo of an adult |
| A photograph of a big object | An image of Andorra |
| Picture taken in Rwanda | Image with a zebra |
| Picture with airplanes | Picture taken in the South African safari |

(b) Specializations of MLP units across layers in OpenCLIP-L.

Figure 11: Comparison of MLP dimensionality and specialization in OpenCLIP-L for ImageNet showing higher dimensionality than head units and strong superposition of concepts. (a) Visualization of MLP dimensionality. (b) Examples of MLP unit specializations across layers.

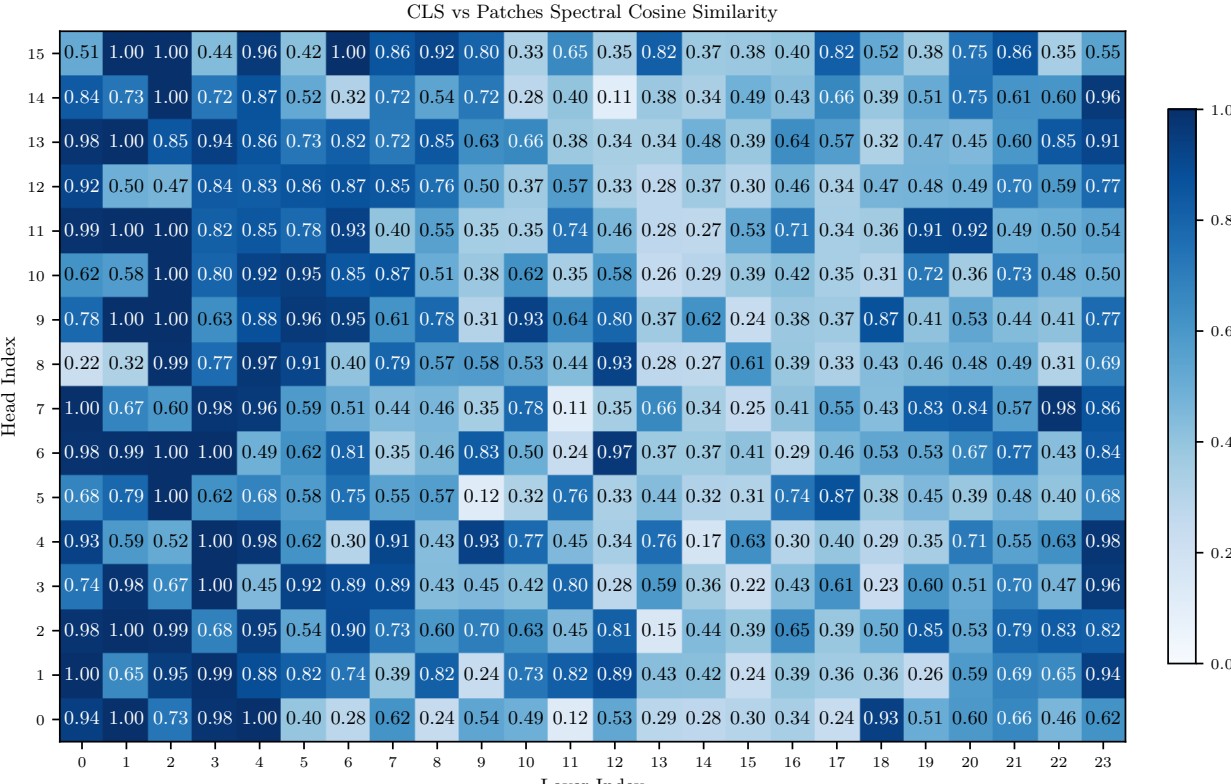

(a)

| L22.H7.PC0 (high similarity) | L22.H11.PC0 (low similarity) |
|---|---|
| Serene winter wonderland | Photo with faded, nostalgic colors |
| A photo taken in the summer | Classic black and white cityscape |
| A photo taken in the winter | Caricature of a cartoon character |
| Image with a dramatic thunderstorm | Nature macro photography |
| Daytime shot | Ocean sunset silhouette |
| **L22.H8.PC0** | **L22.H8.PC1** |
| vast natural landscape | Sublime and serene mountain lake |
| Futuristic technological concept | Enchanting fantasy world |
| Picture captured in the Egyptian pyramids | Image captured in the Japanese tea gardens |
| An irregular shape | A photo with the letter L |
| italic text | A photo with the letter Y |

(b)

Figure 12: `[CLS]`-patches compatibility analysis. **(a)**: Head-wise Spectral Cosine Similarity (Equation (4)) between `[CLS]` and patch tokens for OpenCLIP-L on ImageNet. **(b)**: Specialization of patch tokens on selected heads using OMP applied to specific principal components. L22H8, identified as a "letter" head (Figure 3), exhibits the lowest similarity in the final four layers. Interestingly, pattern-like descriptions emerge in the first principal component (PC0), while letter descriptions become apparent only in the second component (PC1).

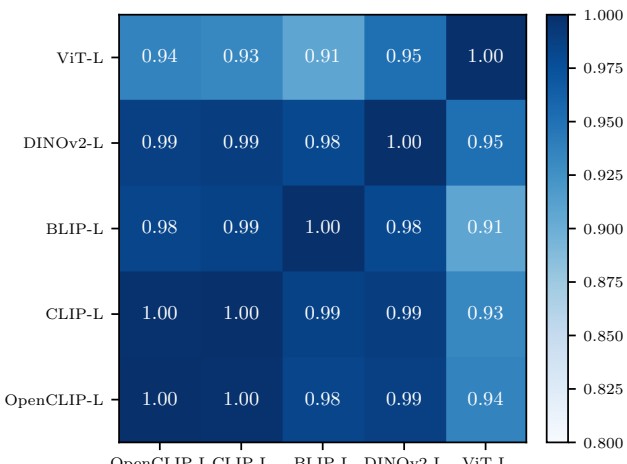

Figure 13: Pearson correlation between cross-dataset similarities on different models. Comparison is done between ImageNet and each of the other datasets and averaged over heads.

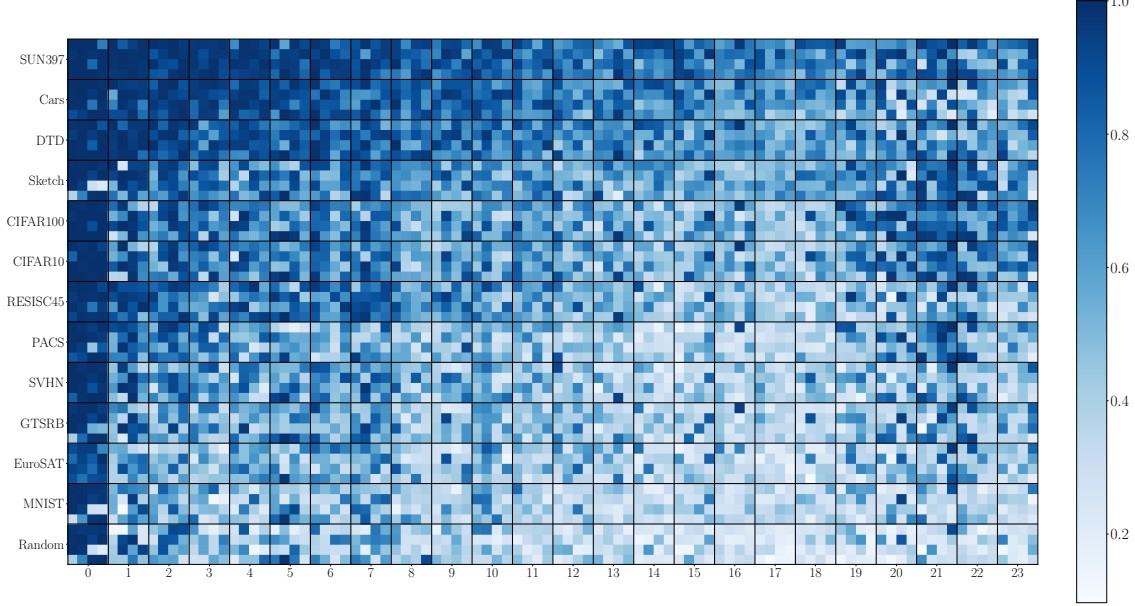

Figure 14: Attention head similarity across layers of BLIP-L, computed between ImageNet head representations and those obtained on other datasets.

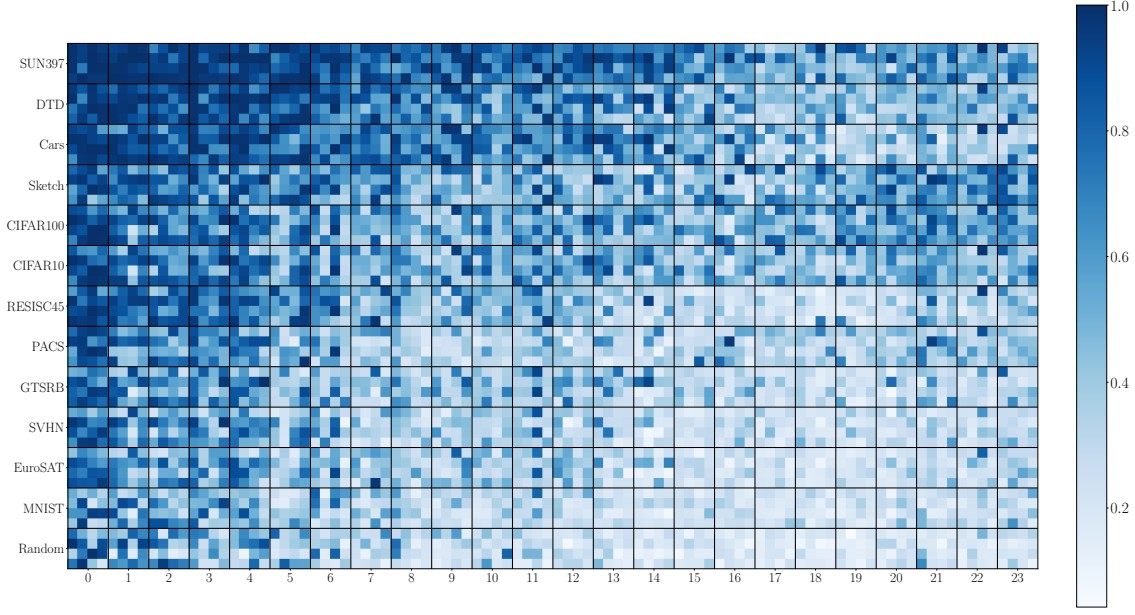

Figure 15: Attention head similarity across layers of CLIP-L, computed between ImageNet head representations and those obtained on other datasets.

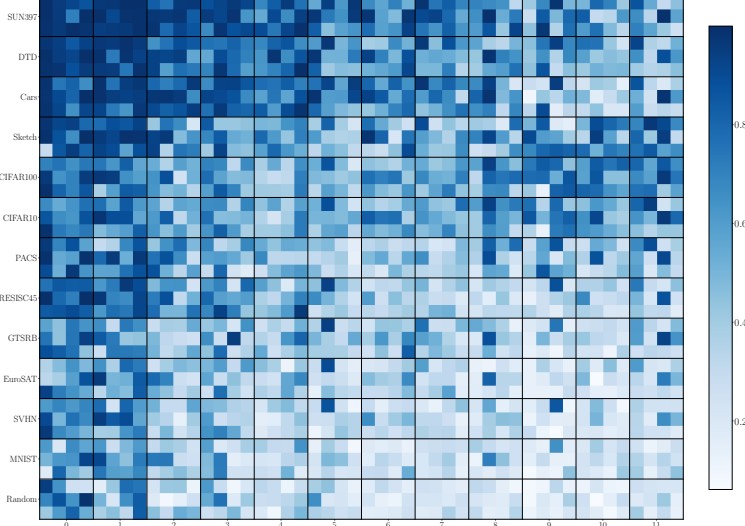

Figure 16: Attention head similarity across layers of CLIP-B, computed between ImageNet head representations and those obtained on other datasets.

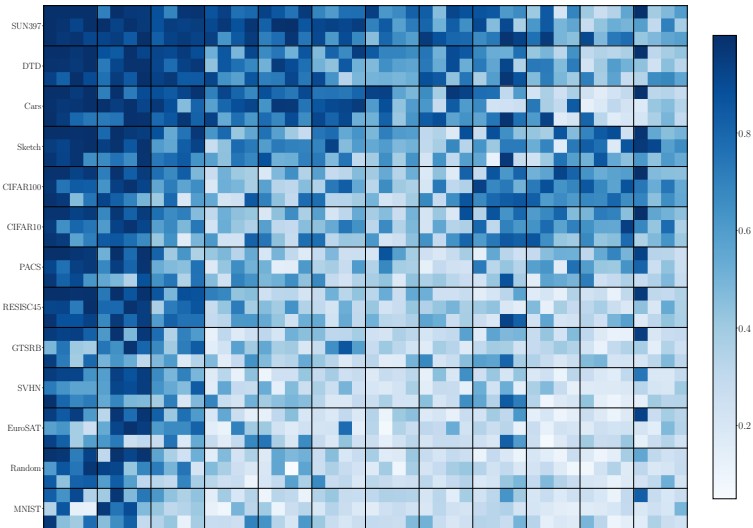

Figure 17: Attention head similarity across layers of OpenCLIP-B, computed between ImageNet head representations and those obtained on other datasets.

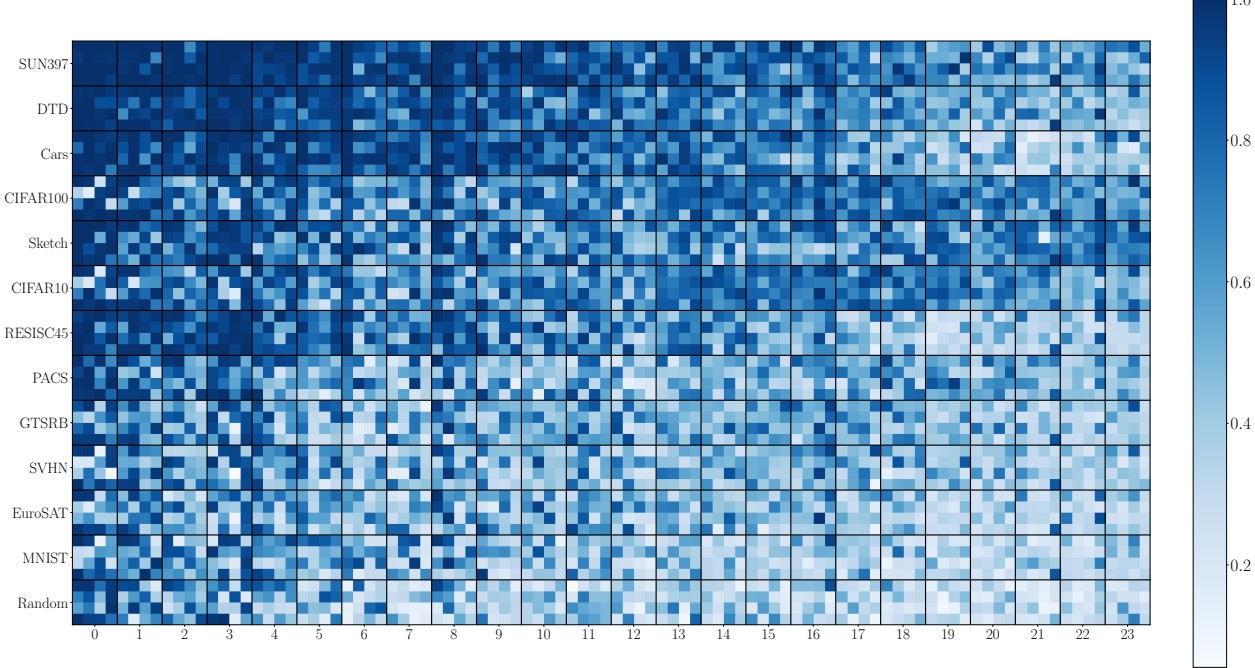

Figure 18: Attention head similarity across layers of DINOv2-L, computed between ImageNet head representations and those obtained on other datasets.

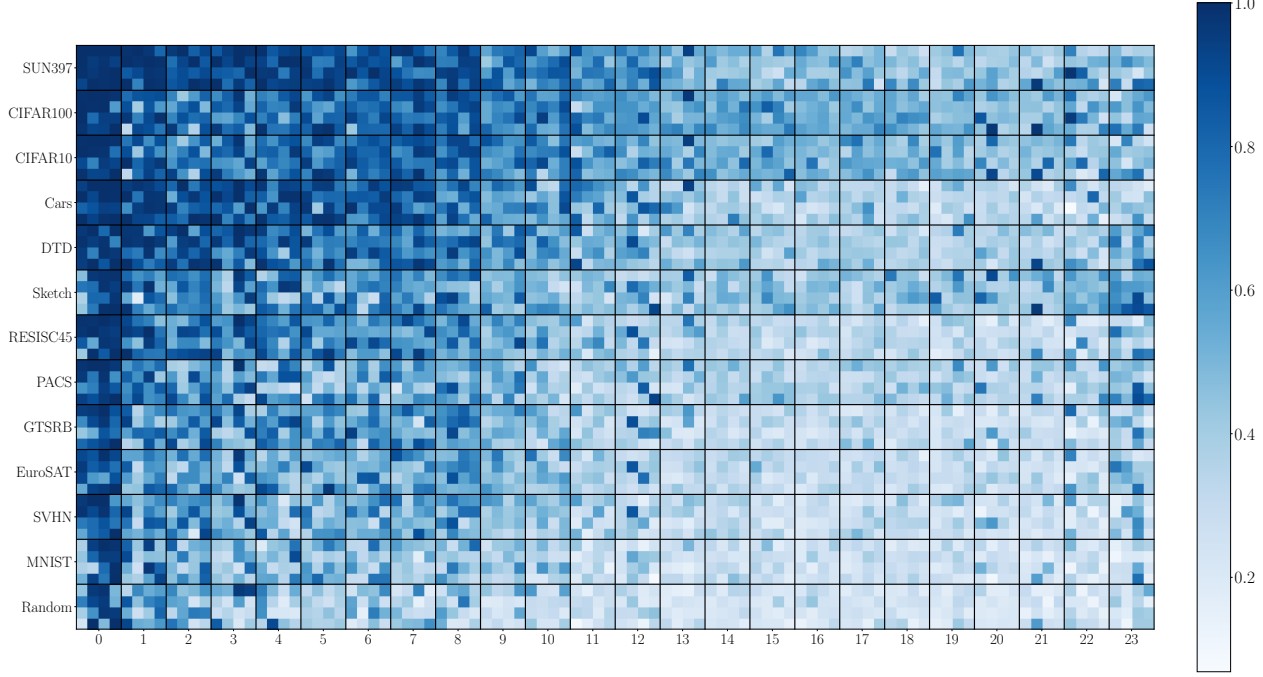

Figure 19: Attention head similarity across layers of ViT-L, computed between ImageNet head representations and those obtained on other datasets.

| Dataset | CLIP-B | | | | | | |
|---|---|---|---|---|---|---|---|
| | U | U\|T | S | R | H | B | O |
| CIFAR10 | 0.87 | 0.90 | 0.89 | 0.41 | 0.81 | 0.91 | 0.93 |
| CIFAR100 | 0.60 | 0.60 | 0.60 | 0.20 | 0.57 | 0.66 | 0.69 |
| Cars | 0.56 | 0.57 | 0.58 | 0.15 | 0.51 | 0.65 | 0.63 |
| DTD | 0.42 | 0.41 | 0.41 | 0.16 | 0.41 | 0.45 | 0.47 |
| EuroSAT | 0.54 | 0.54 | 0.58 | 0.23 | 0.53 | 0.55 | 0.88 |
| GTSRB | 0.33 | 0.34 | 0.40 | 0.14 | 0.25 | 0.42 | 0.61 |
| MNIST | 0.55 | 0.53 | 0.50 | 0.22 | 0.29 | 0.52 | 0.88 |
| RESISC45 | 0.61 | 0.61 | 0.61 | 0.20 | 0.58 | 0.66 | 0.73 |
| SUN397 | 0.42 | 0.42 | 0.47 | 0.12 | 0.42 | 0.64 | 0.64 |
| SVHN | 0.48 | 0.44 | 0.52 | 0.22 | 0.42 | 0.52 | 0.58 |
| Average | 0.54 | 0.54 | **0.56** | 0.21 | 0.48 | 0.60 | 0.70 |

Table 4: Accuracy when doing **zero ablation of all units except top 5%** of attention heads. Heads are assigned a binary weight using an Unsupervised (U), Task-conditioned Unsupervised (U|T), Supervised (S), and Random (R) strategy (a mean over 10 different seeds). H corresponds to using all the attention heads available, B is the original model performance, and O is the optimized continuous weighting case.

| | OpenCLIP-B | | | | | | |
|---|---|---|---|---|---|---|---|
| **Dataset** | U | U\|T | S | R | H | B | O |
| CIFAR10 | 0.94 | 0.94 | 0.94 | 0.49 | 0.92 | 0.95 | 0.96 |
| CIFAR100 | 0.73 | 0.71 | 0.71 | 0.29 | 0.69 | 0.76 | 0.77 |
| Cars | 0.79 | 0.84 | 0.82 | 0.24 | 0.73 | 0.88 | 0.86 |
| DTD | 0.52 | 0.51 | 0.51 | 0.25 | 0.51 | 0.57 | 0.58 |
| EuroSAT | 0.51 | 0.49 | 0.49 | 0.26 | 0.44 | 0.52 | 0.87 |
| GTSRB | 0.49 | 0.47 | 0.48 | 0.12 | 0.42 | 0.50 | 0.66 |
| MNIST | 0.75 | 0.71 | 0.74 | 0.19 | 0.45 | 0.66 | 0.91 |
| RESISC45 | 0.64 | 0.63 | 0.63 | 0.25 | 0.56 | 0.68 | 0.77 |
| SUN397 | 0.55 | 0.55 | 0.56 | 0.20 | 0.46 | 0.70 | 0.69 |
| SVHN | 0.62 | 0.62 | 0.61 | 0.21 | 0.44 | 0.50 | 0.66 |
| Average | **0.65** | **0.65** | **0.65** | 0.25 | 0.56 | 0.67 | 0.77 |

Table 5: Accuracy when doing **zero ablation of all units except top 5%** of attention heads. Heads are assigned a binary weight using an Unsupervised (U), Task-conditioned Unsupervised (U|T), Supervised (S), and Random (R) strategy (a mean over 10 different seeds). H corresponds to using all the attention heads available, B is the original model performance, and O is the optimized continuous weighting case.

| | Method | Atom 0 | Atom 1 | Atom 2 |
|---|---|---|---|---|
| **GTSRB** | U | An acute triangle | The number thirty | Image with a single road sign |
| | U\|T | An acute triangle | The number thirty | Image with a single road sign |
| | S | An acute triangle | The number thirty | A concentric circle |
| | H | An acute triangle | The number thirty | Image with a single road sign |
| | O | An acute triangle | An image of the number 0 | The number thirty |
| | B | An acute triangle | The number thirty | Serene beach sunset |
| **MNIST** | U | Image with three people | Image with a six people | A pendulum |
| | U\|T | Image with three people | Image with a seven people | Image with a pair of subjects |
| | S | An image of three subjects | Image with six subjects | An image of two subjects |
| | H | An image of the number 3 | A pendulum | Image with a seven people |
| | O | An image of three subjects | Aerial view of a coral reef | detailed macro shot |
| | B | Image with a whirlpool of brimstone | An image of the number 3 | Playful siblings |

Table 6: First 3 atoms (decompositions) obtained by SOMP on OpenCLIP-L output encodings after the different coarse unit selection methods are applied.

| | CLIP-L | | | | | | |
|---|---|---|---|---|---|---|---|
| **Dataset** | U | U\|T | S | R | H | B | O |
| CIFAR10 | 0.88 | 0.95 | 0.95 | 0.51 | 0.92 | 0.96 | 0.97 |
| CIFAR100 | 0.72 | 0.73 | 0.72 | 0.32 | 0.68 | 0.75 | 0.80 |
| Cars | 0.74 | 0.74 | 0.73 | 0.29 | 0.70 | 0.78 | 0.79 |
| DTD | 0.50 | 0.50 | 0.51 | 0.26 | 0.51 | 0.55 | 0.61 |
| EuroSAT | 0.71 | 0.71 | 0.73 | 0.36 | 0.53 | 0.62 | 0.95 |
| GTSRB | 0.49 | 0.48 | 0.50 | 0.18 | 0.31 | 0.50 | 0.71 |
| MNIST | 0.78 | 0.77 | 0.85 | 0.36 | 0.75 | 0.76 | 0.96 |
| RESISC45 | 0.63 | 0.63 | 0.64 | 0.30 | 0.59 | 0.71 | 0.84 |
| SUN397 | 0.56 | 0.56 | 0.58 | 0.23 | 0.49 | 0.67 | 0.71 |
| SVHN | 0.65 | 0.65 | 0.65 | 0.30 | 0.60 | 0.58 | 0.71 |
| Average | 0.66 | 0.67 | **0.69** | 0.31 | 0.61 | 0.69 | 0.80 |

Table 7: Accuracy when doing **zero ablation of all units except top 5%** of attention heads. Heads are assigned a binary weight using an Unsupervised (U), Task-conditioned Unsupervised (U|T), Supervised (S), and Random (R) strategy (a mean over 10 different seeds). H corresponds to using all the attention heads available, B is the original model performance, and O is the optimized continuous weighting case.

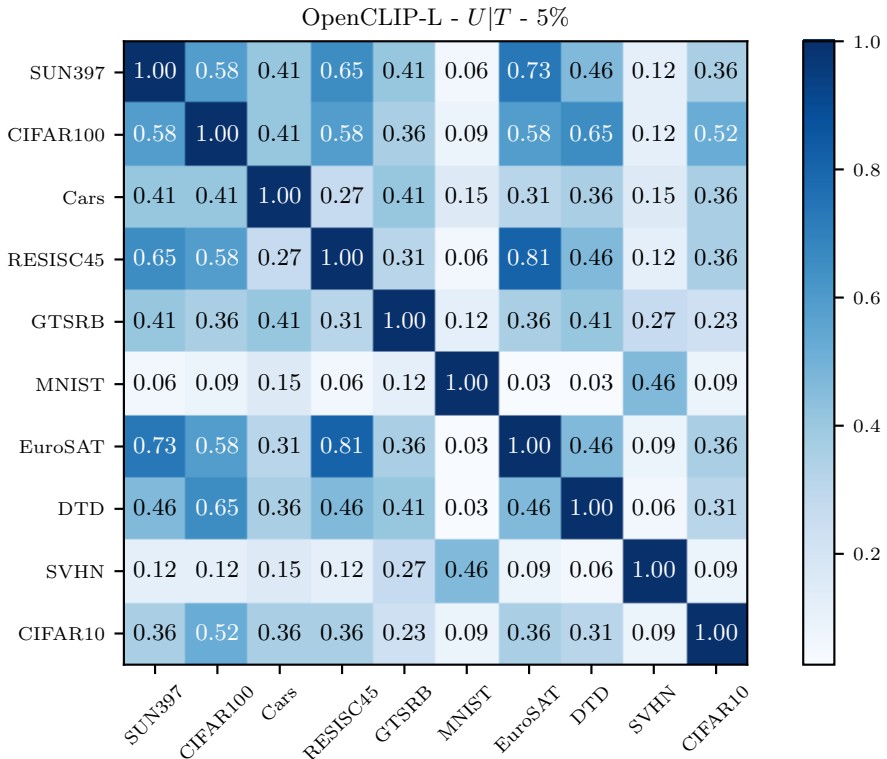

Figure 20: Jaccard similarity between the heads selected by the Task-conditioned Unsupervised method at 5% (introduced in Section 4.1) for each dataset on OpenCLIP-L. It shows meaningful choices, implying that for similar data distributions, similar heads are selected, and their specialization is preserved across datasets (e.g., SVHN, GTSRB, and MNIST have relevant similarities, but also EuroSAT and GTSRB).

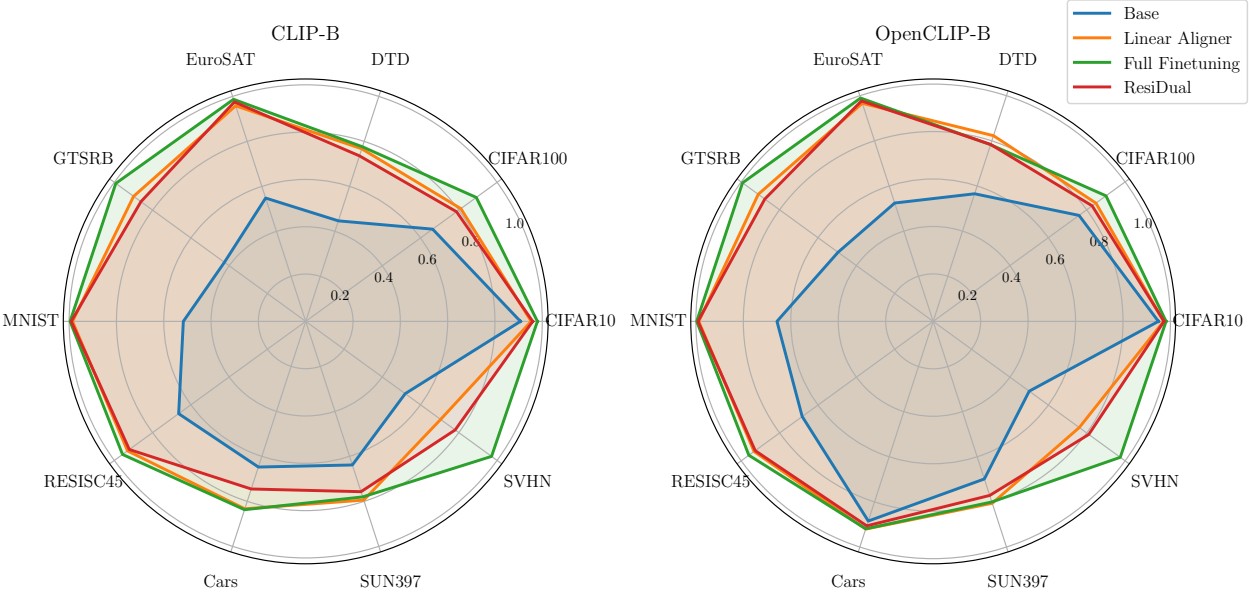

Figure 21: Performance comparison between text-image alignment methods on zero-shot classification tasks.

|          | OpenCLIP-B | | OpenCLIP-L | |
| -------- | -------- | --------- | -------- | --------- |
| **Dataset** | ResiDual | ResiDual* | ResiDual | ResiDual* |
| CIFAR10   | 0.97 | 0.97 | 0.98 | 0.98 |
| CIFAR100  | 0.82 | 0.81 | 0.87 | 0.88 |
| DTD       | 0.78 | 0.75 | 0.84 | 0.81 |
| EuroSAT   | 0.98 | 0.97 | 0.98 | 0.98 |
| GTSRB     | 0.88 | 0.84 | 0.92 | 0.91 |
| MNIST     | 0.99 | 0.99 | 0.99 | 0.99 |
| RESISC45  | 0.93 | 0.91 | 0.95 | 0.95 |
| Cars      | 0.91 | 0.87 | 0.94 | 0.93 |
| SUN397    | 0.77 | 0.72 | 0.82 | 0.80 |
| SVHN      | 0.81 | 0.78 | 0.86 | 0.84 |
| **Average** | **0.88** | 0.86 | **0.92** | 0.91 |

Table 8: Accuracy produced by different configurations of ResiDual using dataset-specific bases and not ImageNet ones as in Section 4.2; (ResiDual): ResiDual in the original formulation (all principal components of all residual units); (ResiDual*): ResiDual limited to head units alone (no MLPs), and PCs truncated to 90% of explained variance.

|          | CLIP-B | | | | OpenCLIP-B | | | |
| -------- | --- | --- | --- | --- | --- | --- | --- | --- |
| **Dataset** | **Lin** | **RD** | **RD*** | **RD$^Y$** | **Lin** | **RD** | **RD*** | **RD$^Y$** |
| CIFAR10   | 0.95 | 0.96 | 0.96 | 0.94 | 0.97 | 0.97 | 0.97 | 0.96 |
| CIFAR100  | 0.81 | 0.79 | 0.76 | 0.71 | 0.85 | 0.83 | 0.82 | 0.78 |
| DTD       | 0.77 | 0.74 | 0.69 | 0.52 | 0.82 | 0.78 | 0.74 | 0.64 |
| EuroSAT   | 0.96 | 0.98 | 0.97 | 0.90 | 0.97 | 0.98 | 0.98 | 0.93 |
| GTSRB     | 0.90 | 0.86 | 0.83 | 0.67 | 0.91 | 0.88 | 0.85 | 0.73 |
| MNIST     | 0.99 | 0.99 | 0.99 | 0.95 | 0.99 | 0.99 | 0.99 | 0.97 |
| RESISC45  | 0.93 | 0.92 | 0.90 | 0.81 | 0.93 | 0.93 | 0.91 | 0.83 |
| Cars      | 0.83 | 0.75 | 0.70 | 0.67 | 0.92 | 0.91 | 0.89 | 0.89 |
| SUN397    | 0.79 | 0.76 | 0.71 | 0.67 | 0.81 | 0.77 | 0.74 | 0.70 |
| SVHN      | 0.71 | 0.78 | 0.74 | 0.60 | 0.76 | 0.81 | 0.79 | 0.68 |
| **Average** | **0.86** | 0.85 | 0.83 | 0.75 | **0.89** | 0.88 | 0.87 | 0.81 |
| **#params** | 262k | 15.4k | 4.9k | 512 | 262k | 15.4k | 4.7k | 512 |

Table 9: Accuracy produced by different configurations of ResiDual, compared with a linear aligner. (Lin): linear aligner at the output level; (RD): ResiDual in the original formulation (all principal components of all residual units); (RD*): ResiDual limited to head units alone (no MLPs), and PCs truncated to 90% of explained variance; (RD$^Y$): ResiDual applied to the output encoding.

