# OpenReview forum: "ResiDual Transformer Alignment with Spectral Decomposition"
_TMLR — Accepted by TMLR_

### Review · Reviewer_SWCe · 2024-12-11

**Summary Of Contributions:**

This paper examines how the residual stream influences specialization in transformer networks, particularly for zero-shot classification tasks. Through systematic analysis, the authors first investigate the geometric properties of attention head representations in Vision Transformers (ViTs) and characterize their emergent specializations. Their experiments reveal that these specialized representations, encoded in the principal components of attention heads, remain remarkably consistent across different data distributions. Leveraging this observation, the authors develop a metric to identify task-specific units in Vision-Language Models, demonstrating that even a simple greedy selection of these specialized units can outperform the full model output in zero-shot settings. Building on these insights, they propose ResiDual, an algorithm that strategically reweights the principal components of residual units. This approach achieves improved model performance while requiring significantly fewer parameters than traditional fine-tuning methods.

**Audience:**

Yes

**Claims And Evidence:**

Yes

**Requested Changes:**

To further strengthen the technical foundations of this work, I recommend addressing three critical aspects: establishing the relationship between MLP and attention head residuals, analyzing token-level information flow under ResiDual, and examining the preservation of head specialization patterns during optimization. These additions would enhance the soundness of the proposed method.

**Strengths And Weaknesses:**

### Strength
- This paper is well-structured, with clear motivations and systematic empirical investigations. The experiments are methodically designed to validate each observation, leading to insightful discoveries.
- The development of the ResiDual algorithm is well-justified through extensive empirical validation of how attention heads operate in low-dimensional manifolds and how their principal components behave across datasets.
- The paper demonstrates ResiDual's effectiveness through comprehensive quantitative evaluation across multiple vision language models (CLIP, BLIP) and diverse datasets. The algorithm also requires fewer parameters than traditional fine-tuning approaches while maintaining comparable performance.

### Weakness
- There is a disconnect between the empirical analysis and algorithm design. While Section 3.1 thoroughly examines the geometric properties and specialization patterns of attention head representations, it does not extend this analysis to MLP layer residuals. Nevertheless, the proposed ResiDual algorithm operates on all residual components. This gap between the analytical foundation and implementation raises questions about the method's theoretical justification when applied to MLP layers.
- In CLIP-like models for zero-shot classification, while the CLS token is primarily used for final predictions, the paper does not adequately address how ResiDual affects the residual flow through patch tokens and their subsequent influence on CLS token representations.
- The observations in Section 3 examine head specialization in a fully intact network, yet the algorithm effectively down weights certain heads during operation. The superior performance of the optimized (O) versus supervised (S) selection methods suggests that head specialization patterns may change under scaling. Without visualization or analysis of how specialization is preserved, this aspect remains unclear.
- The scope of the proposed method is relatively constrained, focusing specifically on zero-shot classification in CLIP-like models. While this represents an important use case, the limited exploration of broader applications leaves questions about the method's generalizability.

---

> ### Author Response · Authors · 2025-01-17
>
> We thank the reviewer for their feedback on our work, which led to interesting new experiments, and we address here their concerns and suggestions:
>
> #### **ResiDual Beyond CLIP-like Models**
>
> We defined a new experimental setting for applying our framework outside CLIP-like models to DinoV2 and ViT. The key ingredient to generalize our alignment experiments is the availability of a non-optimized task against which to test the alignment.
>
> For these encoders, instead of deriving it from text encodings, we use a prototypical approach [a]: for each dataset/task, we average the latent representations of all the corresponding samples for each target class. This outputs representations that contain the average and high-level information for that class. Much like a textual encoding of a simple caption, “An image of <class>”.
>
> The corresponding results are in the revised text in Section A.3 in the Appendix and show that even for these unimodal transformers, some units/components are better aligned with these high-level class prototypes than the output itself.
>
> #### **MLP-Head Unit Relationship**
>
> In the revised text, we added a passage to better contextualize the analysis of Section 3 and why we focus mostly on heads:
>
> > In this paper, we focus on the geometry of attention head representations, as prior works (e.g., [a, b]) have demonstrated their specialization in distinct tasks. MLP layers mix outputs from multiple heads by design, leading to more entangled and less interpretable representations.
> >
>
> We added new results for MLP representations in the appendix. On the *quantitative* side (Figure 11a), they show that the MLP dimensionality over layers follows a similar hunchback trend to that of attention heads. As expected, MLP representations are higher-dimensional. On the *qualitative* side (Figure 11b), we observe “entanglement” of properties in the MLP representations, consistent with the view of MLPs as mostly head mixers.
>
> The minimal contribution of MLPs to the residual specialization becomes more evident in Section 4, where the full ResiDual (which includes the MLP components) yields only marginally better results than ResiDual* (its counterpart using only head units and a subset of their principal components).
>
> #### **Token-level Information Analysis**
>
> To strengthen the paper’s scope and analysis on this aspect, we added an experiment in the revised version. The results are in Fig. 12 in the appendix:
>
> *We analyzed the spectral similarity of principal components computed only on the CLS token for each head (the ones already used as bases in all other experiments) and the ones computed on all the image patch tokens, thus excluding the CLS.*
>
> The quantitative results (Figure 12a) suggest good compatibility overall (0.61 on average -- std 0.23), hinting at principal components of the CLS being compatible across all the patches. We attribute the lower similarities to heads focusing on properties that can vary significantly locally (e.g., patch colors can be completely different even within the same image). We applied Orthogonal Matching Pursuit on the first principal component for the CLS representations and its counterpart across patches to get a qualitative confirmation of this claim (Figure 12b).
>
> The results suggest that although the components vary, they represent the same “semantic” domain and are compatible. This opens the way for the application of ResiDual to be used on more fine-grained downstream tasks (e.g., object segmentation and detection).
>
> #### **Specialization During Optimization**
> We rewrote the introduction to Section 4.1 (Coarse Unit Alignment) to clarify that the selection/optimization methods have no impact on head specialization.
>
> However, all the methods employed in Tab. 1 to filter/reweight whole units target the alignment of the model at the *output level* (i.e., after they are summed) with the classifier, as witnessed by substantial changes in performance. Therefore, we added qualitative explanations obtained by applying Matching Pursuit (SOMP) at the output level for each method in Tab. 6 (appendix) in the revision.
>
> #### **References**
>
> [a] *Analyzing multi-head self-attention: Specialized heads do the heavy lifting, the rest can be pruned. Elena Voita, David Talbot, Fedor Moiseev, Rico Sennrich, and Ivan Titov;* ACL 2019.
>
> [b] *Interpreting clip’s image representation via text-based decomposition; Yossi Gandelsman, Alexei A Efros, and Jacob Steinhardt.* ICLR 2024.

---

### Review · Reviewer_qKGp · 2024-12-20

**Summary Of Contributions:**

This paper studies the geometric and spectral properties of attention head representations in vision transformers (ViTs), focusing on the specialized roles that certain heads play in multimodal (vision-language) models such as CLIP-like architectures. The authors show that these specialized attention heads reside in low-dimensional manifolds and can be characterized via principal components. Building on these insights, they introduce ResiDual, a spectral reweighting technique applied to the residual stream that aims to filter out noise components.

**Audience:**

Yes

**Broader Impact Concerns:**

1.	Amplifying certain principal components may reinforce socially or culturally biased features. The authors should note this risk and suggest bias-screening measures.

2.	Boosting particular components could narrow the model’s focus and overshadow other meaningful attributes, limiting semantic diversity.

**Claims And Evidence:**

Yes

**Requested Changes:**

1.	Clearly define how zero-shot classification is evaluated when language embeddings are not present (e.g., in DINOv2).

2.	Provide ablations showing how changes in dataset composition and complexity affect both intrinsic dimensionality and specialization.

**Strengths And Weaknesses:**

Strengths:

1.	The paper provides a fresh perspective on attention heads in vision transformers by analyzing their low-dimensional spectral structure.

2.	It offers a more interpretable view of how certain heads align with semantic directions, particularly in vision-language settings.

3.	The definitions and theoretical discussions about specific ViTs are thorough and clearly presented, making the core concepts easy to follow.

Weaknesses:

1.	The benefits and interpretation of specialized heads, as well as the impact of ResiDual, are unclear in pure vision models (e.g., DINOv2 [1]) where no language input exists. It would help to clarify whether the method influences representation quality at the image or pixel level.

2.	The zero-shot evaluation setup requires more explicit definitions and baselines. While it fits CLIP-like models, the approach for DINOv2 is less straightforward and should be clearly explained.

3.	The intrinsic dimensionality and specialization may depend heavily on the chosen dataset and tasks. Additional ablations on different datasets and conditions are needed to substantiate the claim that ResiDual generalizes well.

4.	What are considered “noise” or “task-irrelevant” components in both models and downstream tasks? More clarity is needed.

5.	Comparisons and statistical analyses are required to isolate the effect of spectral reweighting. It is unclear how much of the performance gain is directly attributable to the proposed method versus other factors.

[1] Maxime Oquab et al. DINOv2: Learning Robust Visual Features without Supervision. Trans. Mach. Learn. Res. 2024.

---

> ### Author Response · Authors · 2025-01-17
>
> We appreciate the reviewer’s thoughtful feedback, which has helped us refine and expand our work. Below, we provide detailed responses to their concerns and suggestions:
>
> #### **Single Modality Alignment**
>
> In light of the reviewer’s interest in the task alignment of unimodal models, we defined a new experimental setting for applying our framework to DinoV2 and ViT. The key ingredient to generalize our alignment experiments is the availability of a non-optimized task against which to test the alignment.
>
> For these encoders, instead of deriving it from text encodings, we use a prototypical approach [a]: for each dataset/task, we average the latent representations of all the corresponding samples for each target class. This outputs representations that contain the average and high-level information for that class. Much like a textual encoding of a simple caption, “An image of <class>”.
>
> The corresponding results are in the revised text in Section A.3 in the Appendix and show that even for these unimodal transformers, some units/components are better aligned with these high-level class prototypes than the output itself.
>
> #### **Bias Risks**
>
> While we acknowledge that identifying a harmful unit or component could theoretically be exploited to intentionally steer a model’s output towards undesirable behaviors, the reverse is equally valid: such components can be effectively removed or attenuated to mitigate their influence on the model’s output [b, c]. More broadly, any unconstrained fine-tuning or training process inherently carries the risk of relying on biased features or narrowing the model’s focus to a specific task. For this reason, we respectfully argue that this concern is not specific to our framework. However, we appreciate the constructive feedback and will consider incorporating approaches similar to FairPCA [d] in future work to enhance fairness and robustness in the more general setting of choosing a reduced subset of units/components to optimize.
>
> #### **ID/Specialization and Data Distribution**
>
> To address the reviewer’s concerns about the generality of our analysis on intrinsic dimensionality (ID) and specialization across different datasets:
>
> - Regarding ID, in the revision, we extended our quantitative analysis of intrinsic dimensionality (Fig. 10 in the appendix), confirming that heads are consistently low-dimensional and follow the same trend of increasing non-linearity across layers.
> - Regarding specialization, we resort to the ImageNet basis for all our experiments since it is more *versatile* (and storage-efficient) to have a *general enough* (as shown in previous works [e, f]) basis compatible with every dataset (as per the results in Fig. 4, spectral similarity patterns across datasets are meaningful).
>
> However, to further support this choice, in the revised text:
>
> - We added a preliminary experiment (revision, Fig. 20 in the appendix) showing the Jaccard similarity between the heads selected by the Supervised method at 5% (introduced in Section 4.1) for each dataset on OpenCLIP-L. It shows semantically meaningful choices, implying that for similar data distributions, similar heads are selected and, therefore, that their specialization is preserved across datasets (e.g., SVHN, GTSRB, and MNIST have relevant similarities. The same applies to EuroSAT and RESISC45).
> - We added some preliminary results restricted to OpenCLIP-B and OpenCLIP-L using a dataset-specific basis for ResiDual and ResiDual* (Tab. 8 in the appendix), and they show little to no difference in performance.
>
> If the reviewer deems this analysis addresses their compatibility concern, we are happy to extend it to all the models and include them in the appendix for the next revision.
>
> #### **Noise and Task-irrelevant Components**
>
> In the revised text (Introduction, right before the “Contributions” paragraph), we better defined the role of different units contributing to the residual stream. It can be summarized as:
>
> > **task-irrelevant**: when removed, the performance has little to no change.
> > **noise**: when removed, performance improves.
> >
>
> #### **Spectral Reweighting Effect**
>
> In all the experiments for section 4.2, we ensured that the experimental conditions remain strictly controlled: the model parameters are frozen, the dataset is unchanged, and the training random seed is fixed, maintaining the same sample order across runs. The only variable introduced is the alignment operation applied to the residual stream. In the ResiDual case, only $\lambda$ is optimized to improve the task performance. This controlled setup isolates the impact of the reweighting operation from other potential confounding factors. If there are additional factors of interest that require examination and we might have overlooked, we are happy to address them.

---

> > ### Author Response · Authors · 2025-01-17
> >
> > #### **References**
> >
> > [a] *Prototypical Networks for Few-shot Learning; Jake Snell, Kevin Swersky, Richard Zemel;* NeurIPS 2017.
> >
> > [b] *Evaluating the Fairness of Discriminative Foundation Models in Computer Vision; Junaid Ali, Matthäus Kleindessner, Florian Wenzel, Kailash Budhathoki, Volkan Cevher, Chris Russell;* AIES 2023.
> >
> > [c] *Are Two Heads the Same as One? Identifying Disparate Treatment in Fair Neural Networks; Michael Lohaus, Matthäus Kleindessner, Krishnaram Kenthapadi, Francesco Locatello, Chris Russell;* NeurIPS 2022.
> >
> > [d] *Efficient fair PCA for fair representation learning; Matthäus Kleindessner, Michele Donini, Chris Russell, Muhammad Bilal Zafar;* PMLR 2023.
> >
> > [e] *Quantifying Structure in CLIP Embeddings: A Statistical Framework for Concept Interpretation; Yossi Gandelsman, Alexei A. Efros, and Jacob Steinhardt;* ICLR 2024.
> >
> > [f] *Decomposing and Interpreting Image Representations via Text in ViTs Beyond CLIP, Sriram Balasubramanian, Samyadeep Basu, Soheil Feizi;* NeurIPS 2024.

---

> > ### Comment · Reviewer_qKGp · 2025-01-20
> >
> > Thanks for the rebuttal.

---

### Review · Reviewer_yxtr · 2025-01-05

**Summary Of Contributions:**

The paper studies how transformer components specialize in specific tasks, introducing ResiDual - a method that improves vision-language model performance by better aligning specialized components between modalities.

**Audience:**

Yes

**Claims And Evidence:**

Yes

**Requested Changes:**

This is overall an interesting paper, which I think does not need much revision. The suggestion is that some theorems on the relationship between head specialization and the low-dimensional nature of head manifolds should be provided.

**Strengths And Weaknesses:**

Strengths:

1. Innovation in geometric analysis of transformer residual streams and their low-dimensional representations
2. ResiDual provides interpretable modal alignment optimization without extensive parameter tuning
3. Comprehensive empirical validation across many benchmarks.

Limitations:

1. Performance improvements mainly demonstrated in zero-shot classification, efficacy in other downstream tasks unclear
2. Lack of theoretical explanations for the head specialization and the low-dimensional nature of head manifolds.

---

> ### Author Response · Authors · 2025-01-17
>
> We thank the reviewer for their constructive feedback and their overall positive assessment of our work.
>
> #### **Theoretical Grounding**
>
> While our work focuses on empirical analysis, it reflects broader principles in identifiability (e.g., [a, b, c, d]) and mechanistic interpretability (e.g., [e, f]) that the community is actively researching. In general, identifiable components often emerge as a byproduct of regularizations or alignment constraints.
>
> Here, we empirically observe a similar phenomenon: head components *emerge* as low-dimensional and consistent across datasets, suggesting they encode stable features. This consistency hints at an identifiable structure implicitly shaped by training objectives, such as alignment in multimodal models and/or regularities in data distributions.
>
> While a full theoretical treatment is beyond the scope of this work, we believe these considerations help frame our findings.
>
> #### **ResiDual Beyond Classification**
>
> In the revision, we added an experiment (Fig. 12 in the appendix) which allows for the application of ResiDual beyond classification tasks: we measure the compatibility between the CLS bases and patch bases across the transformer (OpenCLIP-L) on ImageNet, observing a consistent spectral cosine similarity.  This hints at the possibility of using a shared basis for all the tokens, offering a more general framework for downstream tasks involving token-level information, such as object detection.
>
> #### **References**
>
> [a] *Learning Independent Causal Mechanisms; Giambattista Parascandolo, Niki Kilbertus, Mateo Rojas-Carulla, Bernhard Schölkopf;* PMLR 2018.
>
> [b] *The Incomplete Rosetta Stone Problem: Identifiability Results for Multi-View Nonlinear ICA; Luigi Gresele, Paul K. Rubenstein, Arash Mehrjou, Francesco Locatello, Bernhard Schölkopf;* UAI 2020.
>
> [c] *All or None: Identifiable Linear Properties of Next-token Predictors in Language Modeling; Emanuele Marconato, Sébastien Lachapelle, Sebastian Weichwald, Luigi Gresele;* ArXiv Preprint 2024.
>
> [d] *Challenges in Explaining Representational Similarity through Identifiability; Beatrix Miranda Ginn Nielsen, Luigi Gresele, Andrea Dittadi; NeurIPS,* UniReps Workshop 2024.
>
> [e] *On the Origins of Linear Representations in Large Language Models; Yibo Jiang, Goutham Rajendran, Pradeep Ravikumar, Bryon Aragam, Victor Veitch;* ICML 2024.
>
> [f] *The Geometry of Categorical and Hierarchical Concepts in Large Language Models; Kiho Park, Yo Joong Choe, Yibo Jiang, Victor Veitch;* ICML, Mechanistic Interpretability Workshop 2024.

---

### Author Response · Authors · 2025-01-17
**Change List**

We sincerely appreciate the reviewers' valuable feedback on our work. Here, we provide a summary of the resulting revisions, which we believe have significantly improved the quality and clarity of the paper:

**Writing changes**:

- We clarified our definition of 'task-irrelevant' and 'noise' units in the Introduction;
- In Section 3 (The Geometry of Residual Units), we added a passage to clarify the reasons why we mainly focus on head units;
- We added a passage at the beginning of Section 4.1 to better frame the Coarse Unit Alignment experiment and clarify why it does not impact head specialization;
- Revised the future work section to mention the new experiments on unimodal models and the potential applications to model fairness.

**New experiments**:

- In Fig. 10 in the appendix, we analyzed the intrinsic dimensionality of other datasets (complementing the original Fig. 2 in the main text, which is ImageNet-specific);
- In the appendix (Fig. 11), we reported additional results regarding the intrinsic dimensionality and lack of specialization of MLP representations;
- Quantitatively (Fig. 12a) and qualitatively (Fig. 12b) analyzed spectral compatibility between CLS-token and patch-token representations;
- To complement the quantitative results of Tab. 1, we added qualitative explanations of the outputs of different coarse selection methods in Tab. 6;
- Added experiments on unimodal vision encoders (VIT and DINO) to analyze their alignment with prototypical classifiers. We study the unit alignment in Tab.3 and show ResiDual results in Fig. 7;
- Trained ResiDual with dataset-specific bases (Tab. 8 in the appendix) instead of the general basis computed on ImageNet.
- Reported Jaccard similarity results for task-conditioned unsupervised coarse unit alignment across datasets to assess specialization generalization (Appendix, Fig. 20).

---

### Decision · Action_Editor_dkMi · 2025-03-06

**Recommendation:** Accept as is

**Comment:**

All the reviewers agreed that the paper is well-written and has provided comprehensive empirical investigations. The main concern, raised by reviewers SWCe and qKGp, is that the scope of the empirical observations found in this paper is limited to CLIP-like models under zero-shot classification settings. In the rebuttal, the authors have provided a new experiment in the appendix where unimodal vision encoders, such as ViT and DINO, are used, and the results show that the proposed method is effective in these settings as well. The reviewers have acknowledged this new experiment and are satisfied with the authors' response. I recommend accepting the paper as is.

**Audience:**

Empirical understanding of the attention features in transformers is an important topic in the field of deep learning. The paper is likely to be of interest to researchers working on transformers and related areas.

**Claims And Evidence:**

This paper studies the spectral properties of attention head representations in vision transformers (ViTs). The main empirical observation is that certain attention heads are specialized for different tasks and can be characterized by the principal components of the attention head representations. The authors propose a spectral reweighting technique, ResiDual, to filter out noisy components in the residual stream. The authors show that this approach achieves improved model performance using fewer parameters.